# CAN LLMS SOLVE LONGER MATH WORD PROBLEMS BETTER?

**Xin Xu**[1*]**, Tong Xiao**[2*]**, Zitong Chao**[1]**, Zhenya Huang**[2†]**, Can Yang**[1†]**, Yang Wang**[1]

[1]The Hong Kong University of Science and Technology
[2]University of Science and Technology of China
`{xxuca, zchaoaa}@connect.ust.hk, {macyang, wangyang}@ust.hk`
`tongxiao2002@mail.ustc.edu.cn, huangzhy@ustc.edu.cn`
https://github.com/XinXU-USTC/CoLeG-Math

## ABSTRACT

Math Word Problems (MWPs) play a vital role in assessing the capabilities of Large Language Models (LLMs), yet current research primarily focuses on questions with concise contexts. The impact of longer contexts on mathematical reasoning remains under-explored. This study pioneers the investigation of **Co**ntext **Le**ngth **G**eneralizability (**CoLeG**), which refers to the ability of LLMs to solve MWPs with extended narratives. We introduce Extended Grade-School Math (E-GSM), a collection of MWPs featuring lengthy narratives, and propose two novel metrics to evaluate the efficacy and resilience of LLMs in tackling these problems. Our analysis of existing zero-shot prompting techniques with proprietary LLMs along with open-source LLMs reveals a general deficiency in CoLeG. To alleviate these issues, we propose tailored approaches for different categories of LLMs. For proprietary LLMs, we introduce a new instructional prompt designed to mitigate the impact of long contexts. For open-source LLMs, we develop a novel auxiliary task for fine-tuning to enhance CoLeG. Our comprehensive results demonstrate the effectiveness of our proposed methods, showing improved performance on E-GSM. Additionally, we conduct an in-depth analysis to differentiate the effects of semantic understanding and reasoning efficacy, showing that our methods improves the latter. We also establish the generalizability of our methods across several other MWP benchmarks. Our findings highlight the limitations of current LLMs and offer practical solutions correspondingly, paving the way for further exploration of model generalizability and training methodologies.

## 1 INTRODUCTION

Math word problems (MWPs) (Bobrow et al., 1964) are mathematical questions presented in natural language, demanding delicate reasoning for solving. With the flourish of large language models (LLMs) (Brown et al., 2020; Touvron et al., 2023a; OpenAI, 2023; Jiang et al., 2024a), the math reasoning ability measured on MWPs benchmarks (Cobbe et al., 2021; Hendrycks et al., 2021; Xu et al., 2025c) has emerged as a critical evaluation metric to assess the overall capability of these models. The representative benchmarks for MWP, including GSM8K (Cobbe et al., 2021) are typically characterized by concise descriptions in a few sentences, whereas the performance of LLMs in solving longer math problems remains under-explored. (Boaler, 1993) suggests that extensive contexts might hinder rather than facilitate the mathematical reasoning process. This observation raises an important question: Do LLMs exhibit a performance degradation in solving long MWPs? If so, how can we improve the solving accuracy of LLMs on these long MWPs, and what underlying factors contribute to the improvement?

---

\* Equal contribution.
† Corresponding author.

To answer these questions, the chain-of-thought (Wei et al., 2022) (CoT) is conducted on the GSM8K benchmark and questions are segregated based on the accuracy of CoT predictions. Rigid statistical hypothesis testing has revealed significant evidence suggesting that LLMs exhibit decreased performance on MWPs with long context (see Section 2.1). In order to further investigate how different LLMs are affected by lengthy contexts of MWPs, we construct the Extended Grade-School Math (E-GSM), a benchmark comprising MWPs with extended context derived from GSM8K, and two novel evaluation metrics. E-GSM maintains minimal alterations to the conditions and the order of conditions while extending the contextual information of the original problem. We investigate four proprietary and four open-source LLMs, along with three state-of-the-art zero-shot prompts on E-GSM. Our results indicate that the **Co**ntext **Le**ngth **G**eneralization (CoLeG) of these LLMs, the ability for LLMs to do math reasoning in a long context, is limited, particularly with longer MWPs.

To alleviate this issue, we propose two different strategies for proprietary and open-source LLMs to improve the problem solving accuracy, respectively. For proprietary LLMs, inspired by cognitive load theory (Sweller et al., 1998), we develop **Co**ndition-**Re**trieving Instruction (CoRe) prompting technique, which encourages LLMs to retrieve problem conditions first and then apply zero-shot-CoT (Kojima et al., 2022). For open-source LLMs, we suggest incorporating *extension* as an auxiliary task during fine-tuning. We also examine the underlying factors contributing to accuracy improvements by analyzing informativeness and missing step metrics derived from (Golovneva et al., 2023). The experiment results suggest that the primary reason for this enhancement lies in improved reasoning pathways. Furthermore, We validate our strategies on several other MWP benchmarks such as MAWPS (Koncel-Kedziorski et al., 2016), SVAMP (Patel et al., 2021), and GSM-IC (Shi et al., 2023), demonstrating their effectiveness and generalizability.

We summarize our main contributions as follows:

1. We construct the E-GSM dataset, comprising MWPs with longer contexts. Comprehensive experiments on both proprietary and open-source LLMs reveal that math reasoning abilities of LLMs are significantly affected by long context.

2. We develop a new instructional prompt named CoRe for proprietary LLMs , which can improve CoLeG and problem solving accuracy of LLMs on E-GSM.

3. We propose to use *extension* as an auxiliary task to fine-tune open source LLMs and release our fine-tuning dataset comprising 65K CoT data.

4. CoRe and *extension* have demonstrated their strong generalization on several MWP benchmarks.

Our comprehensive evaluation reveals that long MWPs significantly degrade the mathematical reasoning abilities of LLMs, highlighting the limitations of current models. Furthermore, we offer practical solutions for both proprietary and open-source LLMs, and further experiments demonstrate the effectiveness and generalizability of our proposed methods. Our findings provides valuable insights and directions for future research on model generalizability and training methodologies.

## 2 THE E-GSM DATASET

### 2.1 LLMs STRUGGLE TO ANSWER MATH WORD PROBLEMS WITH LONGER CONTEXT

To explore whether the performance of LLMs in mathematical reasoning is adversely affected by longer textual contexts, similar to human performance, we conducted an experiment on GSM8K using CoT with `GPT-3.5-turbo` as the representative. The experiment employed the 8-shot demonstrations provided by (Wei et al., 2022), and the complete prompt can be found in Appendix D. We can compare the answers generated by LLMs with the ground-truth answers and then divide the examples into two groups based on accuracy: the incorrect answers group $G_o$ and the correct answers group $G_1$.

We want to study whether there exists a significant difference in the problem length, characterized by the number of tokens, between these two categories. We hypothesize that the lengths of the problem descriptions for these two groups are from different distributions, denoted $X$ and $Y$. As illustrated in Figure 1, the distributions between $X$ and $Y$ are quite different. To conduct a rigorous analysis, we apply the one-sided Mann-Whitney test (Mann & Whitney, 1947):

$$H_0: \ P(X < Y) = P(X > Y) \leftrightarrow H_1: \ P(X > Y) > P(X < Y).$$

The results are reported following (Fritz et al., 2012): There is significant evidence indicating that the number of tokens in $G_1$ is less than in $G_0$ with $U = 141565$, $P = 0.0000$, which suggests LLMs perform better on short MWPs than longer ones, similar to human problem solvers.

To address the potential condounding factors, we further explore whether longer problem correlates with increased problem difficulty. We utilize the number of steps required by `GPT-3.5-turbo` to solve the problems as a proxy for problem difficulty (An et al., 2023; Wei et al., 2022), denoted by $S$, a discrete random variable with $k$ levels. We assumed that the number of problem tokens is denoted by $T$. To analyze this relationship, we consider the following linear model:

$$\log T = \beta_0 + \beta_1 S_1 + \beta_2 S_2 + \ldots + \beta_{k-1} S_{k-1} + \varepsilon,$$

where $\beta_i$ is the coefficient corresponding to the difference in the dependent variable associated with the $i$-th level of $S$ to the reference category. The logarithmic transformation of $T$ is to ensure homogeneity of variance, confirmed by a Levene's test (Levene, 1960) for homogeneity of variance ($P = 0.1518$). Subsequently, we proceeded with a contrast test (Neter et al., 1996):

$$H_0 : \ C \cdot \boldsymbol{\beta} = 0 \leftrightarrow H_1 : \ C \cdot \boldsymbol{\beta} \neq 0,$$

where $C$ is the contrast coefficients (Appendix B.6), and $\boldsymbol{\beta} = [\beta_1, ..., \beta_{k-1}]$. The results indicate insufficient statistical evidence to confirm the linear contrast ($F = 1.9158$, $P = 0.1476$), suggesting insufficient evidence that a direct relationship exists between the problem length and its difficulty.

These findings eliminate the potential confounding factor that longer questions are more difficult and compellingly indicate that, similar to human solvers, LLMs may be lost in longer MWPs.

Figure 1: The visual comparison suggests the number of tokens in $G_0$ is larger than $G_1$, with Mann-Whitney U test suggesting the significance of these differences. This implies that LLMs struggle to solve longer MWPs, which is similar to humans.

## 2.2 DATASET CREATION AND QUALITY CONTROL

To conduct a more comprehensive analysis of Co-LeG across a range of LLMs, we have created E-GSM as a testing ground. Our approach leverages `GPT-4-turbo` to extend the original GSM8K benchmark. Specifically, we primarily construct the data that should meet the following three requirements: 1) The context of problems should be longer. 2) The ground-truth answers should remain the same. 3) The conditions and their order should not change. This section will detail the construction process.

Initial trails revealed that generated questions were only slightly longer than their original questions, and `GPT-4-turbo` failed to achieve a specified token length as set out in the instructions. To overcome this issue and facilitate the extension of math problems into more elaborate contexts, we adopted a sequential, iterative strategy. The process commences with the GSM8K (Cobbe et al., 2021) test set.[1] During the $r$-th iteration ($1 \leq r \leq R$, where R is the total number of extension rounds), the $i$-th question from the preceding iteration $(r-1)$, denoted as $q_i^{r-1} \in Q_{r-1}$, where $Q_{r-1}$ denotes the set of extended variants after extension round $r-1$, is extended using 2-shot demonstrations with `GPT-4-turbo`. Following the

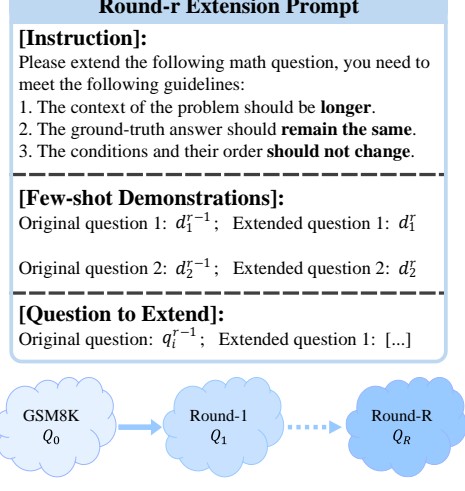

**Round-r Extension Prompt**

[Instruction]:
Please extend the following math question, you need to meet the following guidelines:
1. The context of the problem should be **longer**.
2. The ground-truth answer should **remain the same**.
3. The conditions and their order **should not change**.

- - - - - - - - - - - - - - - - - - - - - -

[Few-shot Demonstrations]:
Original question 1: $d_1^{r-1}$; Extended question 1: $d_1^r$

Original question 2: $d_2^{r-1}$; Extended question 2: $d_2^r$

- - - - - - - - - - - - - - - - - - - - - -

[Question to Extend]:
Original question: $q_i^{r-1}$; Extended question 1: [...]

Figure 2: E-GSM creation process and prompt template for extension.

[1]The test set can be accessed through this link

expansion in the $r$-th round, quality control is performed to ensure quality, resulting in the refined set of extended problems, $Q_r$. The prompt structure, along with the entire extension process, is depicted in Figure 2.[2] As the expansion progresses, we observe a gradual increase in the length of questions. The average tokens for MWPs in each stage are presented in Table 1. In the later phases, a deceleration in the rate of increase in the number of tokens is observed, leading us to end at $R = 4$. As a result, our E-GSM incorporates extended problems from all rounds $\cup_{r=1}^{R} Q_r$. Examples are provided in Appendix A.1.

After obtaining extended questions, we have to ensure they meet our requirements mentioned above. We perform quality control process in the following two steps: 1) We apply human evaluation on a subset of E-GSM to precisely assess the quality of questions. Evaluation results shows that 94.5% questions possess acceptable quality. Human evaluation details are provided in Appendix A.2. 2)

Table 1: The average tokens of E-GSM dataset where the number of tokens is returned by `GPT-4-turbo` and the number of questions in each round of extension.

| Round | $Q_0$ | $Q_1$ | $Q_2$ | $Q_3$ | $Q_4$ |
|---|---|---|---|---|---|
| #Tokens | 77.0 | 192.2 | 301.5 | 363.5 | 385.7 |
| #Questions | 1319 | 1195 | 1143 | 1102 | 1068 |

We devise two heuristics to automatically filter undesired extended questions of the entire dataset. These heuristics are elaborated in Appendix A.3. After the quality control, the number of questions retained in each round is shown in Table 1. Detailed quality control process is given in Appendix A.4).

## 2.3 EVALUATION METRICS ON E-GSM

We propose two metrics to evaluate the performance of LLMs on E-GSM from two different angles: efficacy and robustness. For efficacy, our goal is to check whether a question and all its corresponding variants can be consistently solved, thereby evaluating the model's capability to accurately solve the same question regardless of variations in context length and circumventing potential randomness. For robustness, the relative performance drop of the accuracy from $Q_0$ to $Q_R$ is used, where $Q_r$ is the set of round-r extended questions. For a given question $q$, its ground-truth is denoted by $\text{gt}(q)$, and the answer generated by the method $\mathcal{M}$ is represented as $\mathcal{M}(q)$. The following metrics are considered:

**Round-$r$ accuracy** $\text{Acc}_r(\mathcal{M}) = \text{Acc}(\mathcal{M}; Q_r)$ is defined as the average accuracy of method $\mathcal{M}$ on the set of problems $Q_r$.

$$\text{Acc}(\mathcal{M}; Q_r) = \frac{\sum_{q \in Q_r} \mathbb{I}[\mathcal{M}(q) = \text{gt}(q)]}{|Q_r|},$$

where $\mathbb{I}$ is the indicator function and $|Q_r|$ denotes the number of round-r extended questions (from Table 1).

**CoLeG-E** quantifies the **e**fficacy of **CoLeG**, which is defined as the averaged accuracy of $\mathcal{M}$ to solve a seed question and all its corresponding extension variants.

$$\text{CoLeG-E}(\mathcal{M}) = \frac{\sum_{q_i^R \in Q_R} \left[ \wedge_{r=1}^{R} \mathbb{I}[\mathcal{M}(q_i^r) = \text{gt}(q_i^r)] \right]}{|Q_R|},$$

where $\wedge$ denotes "all" operation and $\mathbb{I}$ is the indicator function. CoLeG-E evaluates the proportion of original problems that can be consistently solved across all levels of extended context.

**CoLeG-R** assesses the **r**obustness facet of **CoLeG**, which is characterized by the relative accuracy drop rate on $Q_R$ compared to the performance on initial questions $Q_0$:

$$\text{CoLeG-R}(\mathcal{M}) = 1 - \frac{\text{Acc}_0(\mathcal{M}) - \text{Acc}_R(\mathcal{M})}{\text{Acc}_0(\mathcal{M})}.$$

In addition, to further investigate the underlying factors behind LLMs' abilities in solving long MWPs, we leverage informativeness and missing step Golovneva et al. (2023) to evaluate the semantic understanding and mathematical reasoning facets of LLMs (will be discussed in Section 4.3). A detailed calculation and explanation of these metrics can be found in Appendix B.2.

---

[2]The full prompts can be found in Appendix D.

**Question:** On a warm Wednesday afternoon, Shannon treats herself to a refreshing iced coffee. Shannon makes herself iced coffee by brewing hot coffee and adding ice cubes. Each ice cube cools the coffee by 13 degrees but makes the coffee 12 milliliters weaker. Any added liquid also weakens the coffee. How many milliliters weaker is Shannon's iced coffee when it is cooled by 65 degrees and she adds 15 milliliters of cream?

Let's think step by step.

**0-CoT**

LLMs →

Step 1: Calculate how many ice cubes Shannon needs to cool the coffee by 65 degrees.
To cool the coffee by 65 degrees, Shannon needs to use 65 / 13 = 5 ice cubes.
Each ice cube makes the coffee 12 milliliters weaker, so 5 ice cubes will make the coffee 5 * 12 = 60 milliliters weaker. ✗

Let's first understand the problem, then list all the known conditions which are formed by numbers or quantitative relationships along with their contexts from problem text, and identify the final goal of the problem. Then let's solve the problem step by step.

**CoRe**

LLMs →

**1. Known Conditions:**
······ Shannon cools the coffee by 65 degrees. Shannon adds 15 milliliters of cream.
**2. Goal:**
Calculate the total weakening effect on Shannon's iced coffee when it is cooled by 65 degrees and 15 milliliters of cream are added.
**3. Solution Steps:**
······ Add the weakening effect of the cream: 60 milliliters + 15 milliliters = 75 milliliters. Shannon's iced coffee becomes 75 milliliters weaker when cooled by 65 degrees and 15 milliliters of cream are added. ✓

Figure 3: A comparison between solving a long problem (shortened version) with 0-CoT and CoRe.

## 3 METHODOLOGY

In this section, we detail our novel approaches aimed at enhancing CoLeG. Given that access to the model weights of proprietary LLMs is restricted, we have developed a new instructional prompt, to increase the abilities of LLMs in solving long MWPs (Section 3.1). For open-source LLMs, we introduce a novel auxiliary task, *extension*, to boost CoLeG by enriching the training data (Section 3.2).

### 3.1 CONDITION-RETRIEVING INSTRUCTION FOR PROPRIETARY LLMs

Although proprietary LLMs exhibit strong math reasoning capability, they are still negatively impacted by long context in solving MWPs (Section 2.1). Our careful analysis of their generated solutions suggests that this issue stems from the LLMs' difficulty in simultaneously processing all the information presented in lengthy problems while performing mathematical reasoning. As a result, they may overlook critical details necessary for solving the problems, ultimately hindering their reasoning abilities.

Cognitive load theory (Sweller et al., 1998) suggests that humans are only conscious of the contents that exist in a limited-size working memory and all other information is hidden until it is swapped into working memory. Drawing this inspiration, we suppose that a similar mechanism exists for LLMs in solving MWPs: Long MWPs make the "LLMs' working memory" saturated by irrelevant contextual details, and missing key conditions that are essential to solve the problem.

To alleviate this issue, we propose a novel **Co**ndition-**Re**trieving instruction (**CoRe**), which divides the problem solving process into two parts. Firstly, CoRe begins by guiding LLMs to identify the conditions and ultimate objective of the given problem, making sure "LLMs' working memory" is filled with essential conditions rather than irrelevant contextual details, and then apply 0-shot CoT Kojima et al. (2022) to prompt LLMs to solve the original problem. As shown in Figure 3, LLMs using 0-shot CoT fail to identify the essential information "she adds 15 milliliters of cream after the coffee is cooled" due to the influence of contextual details during reasoning, which leads to the wrong answer. In contrast, with the assistance of CoRe, LLMs can first focus on parsing the conditions and the ultimate goal of the original problem. This allows them to concentrate on performing mathematical reasoning based solely on the previously identified essential information. By doing so, they can accurately deduce the numerical answer while minimizing the influence of irrelevant contextual details on their reasoning process. For more comparative examples, please refer to Appendix E. Min et al. (2023) proposes a similar approach, suggesting that breaking down information into smaller components can enhance the evaluation of factual precision.

### 3.2 EXTENSION AS AN AUXILIARY TASK FOR OPEN-SOURCE LLMs

We can generate longer MWPs using *extension* technique and apply quality control to filter out extended questions with poor quality introduced in Section 2.2. The new questions, their generated

reasoning paths, their corresponding answers are collected as augmented data:

$$\mathcal{D}_r = \{(q_i^r, e_{ij}^r, a_{ij}^r) : a_{ij}^r = \text{gt}(q_i^r); i = 1, 2, ..., N_r; j = 1, 2, ..., K_i^r\},$$

where $r$ denotes the round of *extension*, $i$ is the index of questions, $e_{ij}^r, a_{ij}^r$ are $j$-th augmented reasoning path and $j$-th corresponding answer for the $i$-th question in $r$-th round of extension $q_i^r$, $\text{gt}(\cdot)$ is the ground-truth answer, $N_r$ is the number of questions left in $r$-th round after quality control, and $K_i^r$ is the number of augmented CoT paths for $q_i^r$.[3]

We fine-tune an LLM (parameterized by $\boldsymbol{\theta}$) on $\mathcal{D} = \mathcal{D}_0 \cup \mathcal{D}_1$ by maximizing the log likelihood of the reasoning path and corresponding answer conditioned on the question. The specific loss we used during supervised fine-tuning is given as follows:

$$\mathcal{L}(\boldsymbol{\theta}) = -\sum\nolimits_{(q,e,a)\in\mathcal{D}} \log \mathbb{P}(e, a \mid q; \boldsymbol{\theta}).$$

## 4 RESULTS AND ANALYSIS

### 4.1 EXPERIMENTAL SETUP

**Prompting Baselines**. For prompting methods, we include 4 mainstream proprietary LLMs: `Claude-3-opus`, `Gemini-pro` (Team et al., 2023), `GPT-3.5-turbo`, `GPT-4o-mini` (OpenAI, 2024). To negate the sensitivity of the choice of few-shot demonstrations on models' performance (Fu et al., 2023; Diao et al., 2023), we explore zero-shot prompting techniques (Shi et al., 2023; Chen et al., 2024), including zero-shot CoT (0-CoT) (Kojima et al., 2022), Plan-and-Solve (PS)(Wang et al., 2023a), and a variant of PS (PS+) (see Appendix D).

**SFT Dataset**. To generate our augmented dataset, We generate five reasoning paths for each question in the training set with `GPT-3.5-turbo` and remove any paths that led to incorrect final answers. We get $\mathcal{D}_0$ that incorporate 38,507 valid CoT data points (we also include the GSM8K training set in $\mathcal{D}_0$) and $\mathcal{D}_1$ that includes 26,422 CoT data for extended quesitons. The entire training set, represented as $\mathcal{D} = \mathcal{D}_0 \cup \mathcal{D}_1$, incorporates 64,929 CoT data. Appendix B.3 shows the input-output formats of the SFT data and examples from both $\mathcal{D}_0$ and $\mathcal{D}_1$. We also collect $\mathcal{D}_2$ of 24,147 CoT data to study the effect of scaling up the SFT data (see Section C.2).

**SFT Baselines**. For fine-tuning open-source LLMs, our study encompasses LLaMA-2 (Touvron et al., 2023b) across different model scales and Mistral-7B (Jiang et al., 2023). More detailed descriptions of these LLMs can be found in Appendix B.1. The baseline for SFT is fine-tuning without *extension* (SFT on $\mathcal{D}_0$). Detailed SFT settings can be found in Appendix B.5.

### 4.2 MAIN RESULTS

We compare the performance of various proprietary LLMs using different zero-shot prompting techniques and open-source LLMs of varying sizes on E-GSM. The main results are summarized in Table 2. Our observations are as follows:

**There is a noticeable trend of diminishing accuracy from one round to the next in all LLMs**, both proprietary and open-source. On average, $\text{Acc}_0$ surpasses $\text{Acc}_4$ by 15.33% when using various 0-CoT prompts for different proprietary LLMs and by 17.41% for open-source LLMs trained without extension across various model sizes and backbones. This trend indicates a decline in performance for LLMs when addressing longer questions, even when the difficulty level remains constant. Notably, `Claude-3-opus` achieves the highest accuracy in all rounds among the four proprietary LLMs evaluated. However, it still exhibits a drop in accuracy of more than 10% from round 0 to round 4, despite using different zero-shot prompts.

**CoLeG-E is significantly lower than $\text{Acc}_i$ and CoLeG-R deviates from 1 for all LLMs**. As discussed in Section 2.3, CoLeG-E measures the percentage of problems accurately solved in all rounds, while $\text{Acc}_0$ denotes the average performance of LLMs in the original GSM8K. The discrepancy between these two indicates that even though LLMs can achieve high accuracy on GSM8K, there are inconsistencies when problems become longer. Specifically, a model may sometimes solve a problem

---

[3]$\mathcal{D}_0$ refers to applying answer augmentation to the original GSM8K training set (see Section 4.1).

Table 2: Main Results (in %) of CoRe and *extension*.

| Technique | CoLeG-E | CoLeG-R | $Acc_0$ | $Acc_1$ | $Acc_2$ | $Acc_3$ | $Acc_4$ |
|---|---|---|---|---|---|---|---|
| | | | Claude-3-opus | | | | |
| PS | 74.81 | 84.07 | **95.45** | 92.30 | 88.28 | 85.30 | 80.24 |
| PS+ | 75.00 | 84.79 | 95.30 | 91.38 | 87.49 | 84.48 | 80.81 |
| 0-CoT | 74.72 | 86.08 | 94.09 | 91.80 | 87.58 | 84.57 | 80.99 |
| CoRe | **77.81** | **86.29** | 95.38 | **92.64** | **88.98** | **85.39** | **82.30** |
| | | | Gemini-Pro | | | | |
| PS | 46.25 | 77.35 | 80.74 | 75.82 | 69.99 | 65.15 | 62.45 |
| PS+ | 48.22 | 79.19 | 80.52 | 76.82 | 70.25 | 66.88 | 63.76 |
| 0-CoT | 49.16 | 75.69 | 83.70 | 77.91 | 71.83 | 66.76 | 63.36 |
| CoRe | **53.65** | **81.44** | **83.70** | **81.26** | **75.48** | **72.78** | **68.16** |
| | | | GPT-3.5-turbo | | | | |
| PS | 41.76 | 80.43 | 78.70 | 76.65 | 70.87 | 67.15 | 63.30 |
| PS+ | 48.03 | 81.14 | 80.67 | 78.83 | 72.70 | 68.87 | 65.45 |
| 0-CoT | 46.63 | 79.63 | 80.89 | 77.91 | 72.53 | 68.24 | 64.42 |
| CoRe | **51.97** | **83.64** | **83.40** | **81.26** | **75.33** | **73.23** | **69.76** |
| | | | GPT-4o-mini | | | | |
| PS | 72.10 | 83.73 | **93.71** | 91.38 | 85.30 | 81.49 | 78.46 |
| PS+ | 73.60 | 85.09 | **93.86** | **91.72** | 87.05 | 82.85 | 79.87 |
| 0-CoT | 71.91 | 83.40 | 93.40 | 91.05 | 86.26 | 81.58 | 77.90 |
| CoRe | **73.78** | **86.02** | 93.18 | 90.79 | **87.14** | **83.30** | **80.15** |
| | | | LLaMA-2-7B | | | | |
| $\mathcal{D}_0$ | 20.22 | 66.64 | 58.45 | 49.62 | 42.96 | 40.93 | 38.95 |
| $\mathcal{D}$ | **28.09** | **80.97** | **59.44** | **57.57** | **50.92** | **49.46** | **48.13** |
| | | | LLaMA-2-13B | | | | |
| $\mathcal{D}_0$ | 32.40 | 73.91 | **67.02** | 63.10 | 56.87 | 51.09 | 49.53 |
| $\mathcal{D}$ | **37.27** | **84.78** | 66.49 | **66.03** | **61.42** | **58.62** | **56.37** |
| | | | LLaMA-2-70B | | | | |
| $\mathcal{D}_0$ | 45.32 | 81.76 | 76.27 | 74.90 | 69.12 | 66.42 | 62.36 |
| $\mathcal{D}$ | **49.81** | **84.57** | **78.17** | **76.23** | **71.30** | **67.15** | **66.10** |
| | | | Mistral-7B | | | | |
| $\mathcal{D}_0$ | 42.13 | 75.19 | 75.59 | 71.80 | 64.39 | 61.25 | 56.84 |
| $\mathcal{D}$ | **48.50** | **83.65** | **76.12** | **74.48** | **69.82** | **66.70** | **63.67** |

successfully but fail when the problem's context is extended. CoLeG-R quantifies the robustness of mathematical reasoning on a macro scale. A CoLeG-R value deviating from 1 signifies a substantial decrease in performance from one round to the next. With the belief that a model capable of "truly" solving a problem should be able to do so regardless of changes in context (Srivastava et al., 2024; Qian et al., 2024; Ahn et al., 2024), there is still considerable room for improvement in developing a model with a higher CoLeG-E and a CoLeG-R value close to 1.

**The accuracy in original GSM8K ($Acc_0$) is insufficient for evaluating LLMs, while CoLeG-E and CoLeG-R offer additional assessment across other dimensions.** Traditional leaderboards evaluate LLMs based solely on accuracy with the original GSM8K, providing only a limited view of model capability and often failing to effectively differentiate between LLMs. For instance, although `Gemini-Pro` achieves a higher $Acc_0$ than `GPT-3.5-turbo` under 0-CoT (83.70% vs. 80.89%), it consistently exhibits lower accuracy in later rounds, leading to a lower CoLeG-R (75.69% vs. 79.63%) compared to `GPT-3.5-turbo`. This suggests that `Gemini-Pro` is less robust when contexts become longer. On the other hand, `Gemini-Pro` has a higher CoLeG-E than `GPT-3.5-turbo` under 0-CoT (49.16% v.s. 46.63%), implying that `Gemini-Pro` is more effective at solving long MWPs but less robust (a lower CoLeG-R). Combining CoLeG-E with CoLeG-R, we can provide more in-depth analysis of the evaluation of LLMs on math reasoning than the original accuracy alone.

**CoRe achieves the best performance on E-GSM across almost all proprietary LLMs compared to zero-shot prompting baselines.** The improvements are particularly evident in higher rounds of E-GSM. Take `GPT-3.5-turbo` for example, CoRe achieves an improvement over 0-CoT by 2.51% in $Acc_0$ and 5.34% in $Acc_4$. Case study (Appendix E) has shown that our approach to solve long MWPs involves initially extracting useful given conditions and the final goal while disregarding unimportant information. This is followed by sophisticated mathematical reasoning based on the information carefully extracted to solve the problem. Furthermore, CoRe has the highest CoLeG-E and CoLeG-R among all proprietary LLMs evaluated, indicating that our method not only enhances the efficacy of solving long MWPs, but also increases robustness when encountering long contexts.

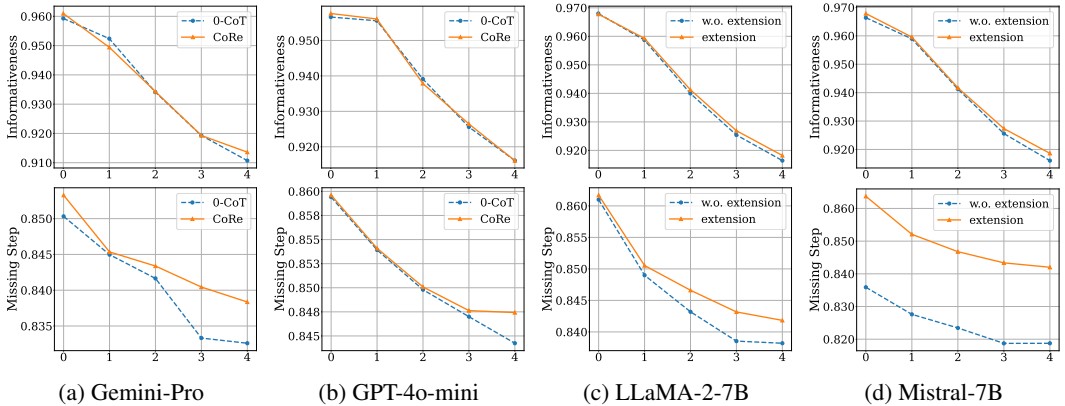

Figure 4: Informativeness and missing step values of 4 representative LLMs

**SFT with *extension* consistently and significantly enhances CoLeG on open-source LLMs**, as reflected in metrics CoLeG-E and CoLeG-R as well as $Acc_i$. On average, SFT with *extension* (on $\mathcal{D}$) leads to an improvement of $5.9\%$ in CoLeG-E and $9.12\%$ in CoLeG-R over SFT without *extension* (on $\mathcal{D}_0$). In particular, for LLaMA-2-7B, there is a substantial absolute enhancement of $7.87\%$ in CoLeG-E and $14.33\%$ in CoLeG-R with our *extension*. This indicates that LLMs fine-tuned with *extension* demonstrate heightened effectiveness in solving MWPs with lengthy contexts and exhibit increased resilience against performance degradation in such a scenario, compared to naive SFT.

### 4.3 FINE-GRAINED ANALYSIS ON SEMANTIC UNDERSTANDING AND MATH REASONING

To deeply understand the underlying factors contributing to the improvements of our methods across various LLMs, we conduct a more fine-grained analysis on semantic understanding and math reasoning abilities of LLMs in solving long MWPs. We leverage informativeness and missing step metrics derived from (Golovneva et al., 2023) to capture these two facets, respectively (see also Section 2.3). The results are presented in Figure 4.

From Figure 4, we observe a performance decline in both informativeness and missing step values across all LLMs from one round to the next, indicating that both their problem understanding and mathematical reasoning abilities are negatively impacted by longer contexts, with a more pronounced drop in understanding, collectively leading to the accuracy degradation in solving longer MWPs.

Furthermore, both CoRe and *extension* enhance the math reasoning ability (shown by the missing step value) of proprietary and open-source LLMs, respectively. The improved math reasoning ability accounts for the improvement of problem solving accuracy in Table 2, which underscores the validity of our motivations and the effectiveness of our methods.

In contrast, we observe that the semantic understanding of both proprietary and open-source LLMs remains generally unchanged after applying CoRe or *extension*, suggesting that neither prompting techniques nor supervised fine-tuning significantly impact the semantic understanding capabilities of LLMs. Based on this observation, we hypothesize that the language understanding ability of LLMs is predominantly established during the pre-training stage and is minimally influenced by post-training adjustments or prompts, which differs from reasoning skills, which can be optimized by employing specific reasoning patterns in prompts or through fine-tuning with additional data.

### 4.4 EXTENSION WITH SPECIALIZED MATHEMATICAL LLMS

In Figure 5 (Left), we have shown the comparison of SFT with *extension* on LLaMA-2 with MetaMath (Yu et al., 2023) across different model sizes. Note that "SFT" in the legend Figure 5 refers to SFT with *extension*. **Interestingly, the accuracy of SFT with *extension* on LLaMA-2 can be comparable to pre-SFT MetaMath, despite the fact that our $\mathcal{D}$ consists of only around 65K CoT data compared to 400K that MetaMath has been trained on.** Furthermore, *extension* can also improve MetaMath in terms of accuracy in all rounds. These observations suggest that extending the questions included

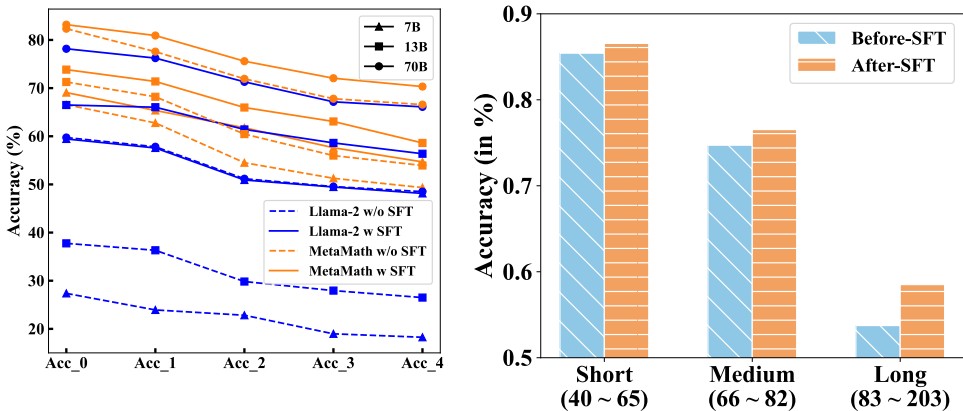

Figure 5: Left: $Acc_i$ varying over rounds in E-GSM of LLaMA-2 and MetaMath families. "w" and "w/o" stand for "with" and "without" respectively. Right: Accuracy on GSM8K with short, medium, and long length. The range of tokens within each category is in parenthesis.

in the training set could serve as an effective data augmentation strategy to improve the mathematical reasoning abilities of LLMs.

***Extension* as an auxiliary task can further improve MetaMath (Yu et al., 2023) in solving longer questions on GSM8K**. While MetaMath outperforms other math-related LLMs on GSM8K, it still faces challenges with lengthy MWPs (Yu et al., 2023). To investigate whether *extension* would help, the GSM8K test set is divided into three subsets of equal size based on question length and the accuracy is calculated over each subset. As shown in Figure 5 (right), *extension* not only benefits questions in all subsets but also yields the most significant performance improvement within the long-length group. Scaling up augmented data of *extension* by adding more rounds of extension to the training set (i.e., getting $\mathcal{D}_r$ for larger $r$) can be beneficial for LLMs to solve long MWPs as well (see Appendix C.2). In addition, We believe that *extension* can be applied orthorgonally to other data augmentation methods provided by (Yu et al., 2023) to further mitigate the challenge of longer MWPs and we leave these for future work.

## 4.5 GENERALIZATION TO OTHER BENCHMARKS

To evaluate generalizability, we assess CoRe and fine-tuned LLMs (without further SFT on corresponding training set) on other benchmarks including MAWPS (Koncel-Kedziorski et al., 2016), SVAMP (Patel et al., 2021), and GSM-IC (Shi et al., 2023). Detailed experimental setups are given in Appendix B.7. The results are shown in Table 3.

**Both our CoRe and *extension* lead to superior performance across all evaluated benchmarks**. These benchmarks consist of MWPs with concise descriptions, making the findings particularly notable: Despite being specifically tailored for long MWPs, these approaches are also effective for shorter MWPs. Remarkably, CoRe results in a $4.5\%$ absolute increase in accuracy on MAWPS with 0-CoT and $4.15\%$ on GSM-IC with PS+. Additionally, without further finetuning on the corresponding training set, our models (SFT on $\mathcal{D}$) yield an average accuracy increase of $15.28\%$ across different model sizes and benchmarks. This demonstrates the generalizability of our approaches in enhancing the performance of LLMs on MWPs.

## 5 RELATED WORK

**Prompting for Mathematical Reasoning**. Mathematical reasoning (Bobrow et al., 1964; Cobbe et al., 2021; Hendrycks et al., 2021) is recognized as a system-2 task (Kahneman, 2011; Bengio et al., 2021), attracting significant attention in the research community. Recent LLM advancements (Brown et al., 2020; OpenAI, 2023; Ouyang et al., 2022; Jiang et al., 2024a) have led to diverse prompting strategies designed to enhance their ability to perform mathematical reasoning (Ahn et al., 2024). Notable among these is the Chain-of-Thought (CoT) prompting (Wei et al., 2022), which has

Table 3: Solving accuracy (in %) of proprietary and open-source LLMs on other MWP benchmarks. The average number of tokens of MAWPS, SVAMP, and GSM-IC are 52, 54, 80, respectively.

| | GPT-3.5-turbo | | | | LLaMA-2 | | | |
|---|---|---|---|---|---|---|---|---|
| Dataset | PS | PS+ | 0-CoT | CoRe | extension | 7B | 13B | 70B |
| MAWPS | 90.77 | 91.78 | 91.40 | **92.67** | ✗ | 71.22 | 74.93 | 87.57 |
| | | | | | ✓ | **72.52** | **78.21** | **87.74** |
| SVAMP | 71.90 | 75.70 | 71.80 | **76.30** | ✗ | 63.40 | 70.30 | 83.10 |
| | | | | | ✓ | **64.90** | **74.00** | **83.80** |
| GSM-IC | 85.38 | 87.48 | 88.35 | **89.60** | ✗ | 62.45 | 74.40 | 84.77 |
| | | | | | ✓ | **66.48** | **76.68** | **85.22** |

significantly improved reasoning capabilities by encouraging the model to generate intermediate steps. While various few-shot techniques such as Re-reading (Xu et al., 2023) and Stepback prompting (Zheng et al., 2023) have been developed, there is a growing interest in methodologies that enable zero-shot reasoning. These methods, such as the two-stage CoT prompting by (Kojima et al., 2022), Plan-and-Solve (PS) prompting (Wang et al., 2023a), and the self-discovery framework by (Zhou et al., 2024), aim to equip LLMs to handle mathematical problems without few-shot demonstrations. Concurrently, Zhong et al. (2024) put forward DUP prompting to extract and answer core problems. Unlike their three-stage prompting, CoRe requires only one additional instruction before 0-CoT.

**CoT Extension**. Extensions to CoT encompass demonstration selection (Zhang et al., 2023; Diao et al., 2023; Fu et al., 2023), advancements in decoding (Wang et al., 2023b; Wang & Zhou, 2024), and developments for more intricate tasks (Yao et al., 2023b;a; Besta et al., 2024; Ding et al., 2023; Xue et al., 2024a;b). Further explorations within this domain have probed areas such as incorrect answer detection (Xu et al., 2024), investigations of failure modes (Berglund et al., 2023; Shi et al., 2023; Chen et al., 2024; Li et al., 2024), other math-related topics (Gao et al., 2023a; Song et al., 2024; Trinh et al., 2024; Yan et al., 2025), and application in broader domains (Jiang et al., 2024b; Xu et al., 2025a; Liu et al., 2025; Gao et al., 2025; Phan et al., 2025). Our work pioneers the investigation of LLMs' CoLeG in mathematical reasoning, employing a zero-shot prompting to isolate the influence of few-shot examplars.

**Specialized LLMs for Mathematics**. Despite general-purpose LLMs, there remains a persistent interest in maximizing the performace of domain-specific LLMs like mathematics, through strategies such as SFT (Yu et al., 2023; Yan et al., 2024) and continued pretraining. Supervised fine-tuning methods, such as WizardMath's (Luo et al., 2023) combination of PPO training and MAmmoTH's (Yue et al., 2023) knowledge distillation of integrating CoT with Program-of-Thought(Chen et al., 2022; Gao et al., 2023b; Xu et al., 2025b). Additionally, MetaMath (Yu et al., 2023) rewrites mathematical questions in multiple formats to augment the training dataset. On the other hand, a distinct strand of research is dedicated to continuing pretraining of base LLMs on extensive mathematical corpora, exemplified by Minerva (Lewkowycz et al., 2022), Llemma (Azerbayev et al., 2023), InternLM-MATH (Ying et al., 2024) and DeepSeekMath (Shao et al., 2024). Our work explores to what extent CoLeG of these specialized mathematical LLMs will be improved by fine-tuning with our newly proposed auxiliary task - *extension*.

# 6 CONCLUSION

Our study explored LLMs' ability to solve longer MWPs, i.e. CoLeG. We introduced a groundbreaking dataset, E-GSM, designed to test CoLeG of LLMs, along with two metrics to evaluate the efficacy and resilience of LLMs in this setting. Our investigation highlighted a notable CoLeG deficiency in existing zero-shot prompting techniques and open-source LLMs. A new instructional prompt, CoRe, and a novel auxiliary task, *extension*, not only significantly strengthen CoLeG but also show superior performance on GSM8K, and generalized well to other MWP benchmarks. By illuminating a previously underexplored aspect of LLMs' reasoning and offering practical solutions to improve it, our work carved out new pathways for using LLMs in complex problem-solving and set the stage for future research on model generalizability and advanced SFT paradigms.

## ACKNOWLEDGMENTS

This work was partially supported by a grant from the Research Grants Council of the Hong Kong Special Administrative Region, China (Project Reference Number: AoE/E-601/24-N), the National Natural Science Foundation of China (No.62477044) and the Fundamental Research Funds for the Central Universities (No.WK2150110038). We would like to thank all reviewers for their helpful suggestions in improving this paper.

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

## A  E-GSM DETAILS

### A.1  EXAMPLES IN E-GSM

In E-GSM dataset, we extend original set of problems from GSM8K to longer ones round-by-round. GSM8K (Cobbe et al., 2021) is a benchmark of grade-school math word problems, containing a training set of 7473 examples, and a test set of 1319 problems. It is under MIT License and can be accessible at https://github.com/openai/grade-school-math. In table 4 , we show one particular original problem in GSM8K and its corresponding extension problems in the first two rounds. In total, our E-GSM consists of around 4.5K MWPs with extensive naratives, divided into four rounds (Table 1). E-GSM will be released under MIT License for future research.

Table 4: Examples from E-GSM.

| $Q_0$ |
|---|
| A mother goes shopping. She buys cocoa at \$4.20, laundry at \$9.45 and a package of pasta at \$1.35. She pays \$20. **How much change does the cashier give back?** |
| $Q_1$ |
| On a bright Saturday morning, a mother decided to take advantage of the weekend sales at her local supermarket. With a shopping list in hand, she navigated through the aisles, picking up items she needed for the week. Among her finds were a rich, dark cocoa powder priced at \$4.20, essential for her famous chocolate cake. Next, she grabbed a bottle of laundry detergent, a necessity for the upcoming week's laundry, priced at \$9.45. Lastly, she couldn't resist adding a package of pasta to her cart, a steal at just \$1.35, perfect for Wednesday night's dinner. After browsing through the aisles and picking up a few more items, she made her way to the cashier. Handing over a crisp \$20 bill to pay for her purchases, she waited for her change. **How much change did the cashier hand back to her after her purchases?** |
| $Q_2$ |
| On a bright and sunny Saturday morning, the local supermarket was bustling with shoppers eager to take advantage of the weekend sales. Among them was a mother, who had meticulously prepared a shopping list to ensure she got everything needed for the upcoming week. As she entered the supermarket, her eyes scanned the aisles for the items on her list. Her first find was a rich, dark cocoa powder, priced at \$4.20, essential for baking her famous chocolate cake that her family adored. Next, she picked up a bottle of laundry detergent, priced at \$9.45, a necessity for tackling the week's laundry pile. Lastly, she spotted a package of pasta, priced at just \$1.35, perfect for the family's Wednesday night dinner. With her cart filled with these items and a few more essentials, she confidently made her way to the cashier. After unloading her cart and watching the cashier scan each item, she handed over a crisp \$20 bill to cover the cost of her purchases. As the cashier processed the transaction, she anticipated the change she would receive, knowing it would be just enough for a small treat for her children on the way home. **How much change did the cashier hand back to her after her purchases?** |

### A.2  HUMAN EVALUATION OF E-GSM

To ensure the quality of E-GSM, we carried out a human evaluation on 50 selected seed questions and their corresponding extension questions in each round. Three well-trained annotators majoring in mathematics, among whom two are graduate students and one is undergraduate, are asked to answer the following questions (Yes/No):

1. Does the order of the question descriptions (i.e., the conditions) remain the same after extension?

2. Are the conditions clear and unambiguous after extension?

3. Is the question solvable after extension (i.e., can an answer be derived from the conditions)?

4. Does the ground-truth answer remain the same after extension?

Then we can categorized these questions into three quality levels according to the answers of the previous questions: excellent, good, and poor.

- **Excellent**. Affirmative responses to all questions.
- **Good**. At least three affirmative responses, including a mandatory "Yes" to Question 3 and Question 4.

- **Poor**. Otherwise.

The inter-annotator agreement, as measured by Cohen's $\kappa$, is given in Table 5. The results indicate almost substantial agreement between the different annotators.

In Appendix A.1, we have shown an example at an excellent quality level. We show two examples of good and poor quality levels in table 6.

Table 5: Cohen's $\kappa$ scores of our human evaluation. 3-class refers to selecting among poor, good, and excellent; 2-class combines good and excellent and selects between poor and non-poor.

|         | Annotator 1, 2 | Annotator 1, 3 | Annotator 2, 3 | Average |
|---------|----------------|----------------|----------------|---------|
| 2-class | 0.7014         | 0.6455         | 0.8616         | 0.7361  |
| 3-class | 0.6307         | 0.5543         | 0.7675         | 0.6508  |

Table 6: Examples of extended problems of different quality. The example in good quality changes the original condition "Each dog takes .5 hours a day to walk and take care of their business" to "He takes each of his 10 dogs on a half-hour walk." The new condition does not include the time of "taking care of their business". The answer to questions 2 is "No". The poor quality problem disregards the original condition that "he only uses his car to go to work." The answer to question 4 is "No". The changes in conditions are highlighted by red.

| An Example Under Good Level |
|---|
| **Original**: John takes care of 10 dogs. Each dog takes .5 hours a day to walk and take care of their business. How many hours a week does he spend taking care of dogs?
**Round-4**: In a picturesque neighborhood, where the gentle whispers of nature blend harmoniously with the serene ambiance, lives a man named John. His love for dogs is not just a hobby but a profound passion that has transformed his spacious backyard into a haven for 10 lively dogs. Each dog, with its unique personality and zest for life, thrives under John's care, enjoying the lush, green expanse as their playground. John, whose heart is as expansive as the open spaces he provides, starts his day with the first light of dawn, ready to embark on a journey of love and dedication with his canine family. John's daily routine is a testament to his unwavering commitment to the well-being of his dogs. He takes each of his 10 dogs on a half-hour walk, exploring the scenic trails that meander through their neighborhood. These walks are more than just exercise; they are adventures filled with exploration, play, and moments of joy, allowing each dog to connect with the essence of nature. For John, these moments are sacred, an opportunity to deepen the bond with his dogs, ensuring their happiness and well-being are always at the forefront. Given John's dedication to providing a fulfilling life for his dogs, how many hours does he devote each week to walking and caring for his beloved companions, ensuring they experience the joy, health, and exercise they deserve? |
| An Example Under Poor Level |
| **Original**: Andy's car fuel efficiency is 10 MPG (miles per gallon). If the current price for regular gas is $3/gallon, how much money is Andy's car consuming per week if he only uses his car to go to work from Monday to Friday and the one-way distance between his home and office is 5 miles?
**Round-1**: Andy, a dedicated employee at a bustling downtown firm, has a daily routine that involves driving his reliable car to work. His car, known for its decent fuel efficiency of 10 miles per gallon (MPG), is his chosen mode of transportation for the 5-mile journey from his cozy suburban home to the high-rise office building where he works. As the week begins on Monday, Andy prepares for his usual commute, aware that the current price for regular gasoline stands at $3 per gallon. Throughout the week, from Monday to Friday, Andy uses his car exclusively for his work commute, making the round trip each day with the intention of maximizing his time and efficiency. Given these circumstances, how much money is Andy's car consuming per week for his work-related travels? |

## A.3   Heuristic for Quality Control

To minimize human labor, we adopt a heuristic approach powered by an entailment model to automatically eliminate undesired examples after extension. Specifically, a score $S_r^i = ENT(q_r^i, q_0^i)$ is calculated to evaluate the informational equivalence between the extended and its corresponding seed questions.

For this purpose, we utilize the SAUMMAC zero-shot model (Laban et al., 2022), an entailment model trained on MNLI (Williams et al., 2018). This model represents a consistency detection technique

in text summarization, employing an off-the-shelf natural language inference model to determine the pairwise entailment score between texts. To detect all potential informational mismatches in any sentence, we aggregate sentence-level entailment with the $min$ operation:

$$ENT(q_r^i, q_0^i) = \min_j \max_k ent(a_k, b_j),$$

where $a_k, b_j$ are sentences of $q_r^i, q_0^i$ respectively, $ent$ represents the probability of entailment, the $min$ is taken over all sentences of $q_0^i$, and $max$ is taken over all sentences of $q_r^i$. A small $S_r^i$ can suggest a potentially unsatisfactory extension, leading to the exclusion of the question $q_r^i$ if $S_r^j < 0.2$, for any $j \le r$. Ultimately, human evaluation (see Appendix A.2) has shown that this approach effectively filters out all unsatisfactory questions.

### A.4 DETAILED QUALITY CONTROL PROCESS

Fifty seed questions are randomly selected from GSM8K and all questions derived from these selected questions are manually inspected (200 in total).Three annotators assess the quality of the extended questions based on specific criteria and assign a quality level to each question (see Appendix A.2). The final quality level is determined through majority voting. The judging criteria for each quality level and several examples are provided in Appendix A.2. Our evaluation finds that 185 of the 200 extended problems are of excellent quality, 4 are good, and 11 are poor.

Due to resource constraints, it is impractical to conduct human evaluations across the entire dataset. Instead, we employ two heuristics to eliminate problematic instances. We guarantee that these methods will effectively detect all substandard examples within our selected questions and then apply them to the full dataset to ensure its quality.

First, we ensure that the extended variant faithfully retains information from the seed question through computing entailment score between the two. For the $i$-th seed question $q_i^0$ and its associated extended variants $q_i^1, ..., q_i^R$, we derive a score $S_i^r = ENT(q_i^r, q_i^0)$ to evaluate the informational equivalence. More specifically, we adopt the out-of-box entailment model (Laban et al., 2022) to calculate $S_i^r$ (see Appendix A.3). A small $S_i^r$ can suggest a potentially unsatisfactory extension, leading to the exclusion of the question $q_i^r$ if $S_i^j < \tau_1$, for any $j \le r$. Obviously, $Q_r \subseteq Q_{r-1}$ after filtration.

Second, we consider an extended variant unsolvable and filter it out if it cannot be addressed by proficient LLMs using various prompting methods. More concretely, `GPT-4o-mini`, `Claude-3-opus`, and `Gemini-Pro` with 4 different prompting methods are used to generate solutions for each question (see Section 4.2), and questions whose accuracy across 12 solutions is less than $\tau_2$ are discarded.

From human evaluation of a subset, we can see that most of our extended questions are of excellent quality. We adjust the thresholds $\tau_1$ and $\tau_2$ based on human evaluation results, successfully filtering out examples of poor quality by setting $\tau_1 = \tau_2 = 0.2$. The number of questions retained in each round is shown in Table 1.

## B EXPERIMENTAL SETUP

### B.1 LLMS

A variety of proprietary LLMs used in our study of zero-shot prompting methods are listed below:

- **GPT-3.5-turbo**: This represents a refined version within the GPT-3.5 series, equipped with an enhanced understanding and generation of both natural language and coding content. We employed the gpt-3.5-turbo-0125 engine for our experiments.
- **GPT-4o** (OpenAI, 2024): GPT-4o is multimodal, and has the same high intelligence as GPT-4 Turbo but is much more efficient.
- **Gemini-Pro**: Developed by Google, this model is touted for its unmatched scalability across a broad spectrum of tasks. Further information about the API is available at https://cloud.google.com/vertex-ai/docs/generative-ai/model-reference/gemini.
- **Claude-3**: This model is released by Anthropic recently, which is claimed to be comparable or even surpass GPT-4. We use `Claude-3-opus-20240229`.

Access to all OpenAI APIs can be found at https://platform.openai.com/docs/models/overview. Furthermore, the creation of our new E-GSM dataset (Section 2.2) utilizes GPT-4-turbo. To acquire SFT data, GPT-3.5-turbo is leveraged to generate CoT chains (Section 4.1). GPT-3.5-turbo is also instrumental in extracting answers from the responses generated by various LLMs (Section B.4).

To explore SFT, our study employs an array of open-source LLMs and specialized mathematical LLMs.

- **LLaMA-2 families**: We engage with models of varying scales—7B, 13B, and 70B—specifically leveraging the instruction-tuned versions. The instruction-tuned LLaMA-2 is accessible at https://huggingface.co/meta-llama/Llama-2-70b-chat-hf. They are under Llama2 License.
- **Mistral-7B**: The model has undergone instruction-tuning and is available at https://huggingface.co/mistralai/Mistral-7B-Instruct-v0.2. It is under Apache License 2.0.
- **MetaMath** (Yu et al., 2023): This approach fine-tunes LLaMA-2 and Mistral-7B on MetaMathQA, a dataset augmented by rewriting original training questions from multiple perspectives. We also incorporate MetaMath-Llemma. All models are available at https://huggingface.co/meta-math. The license for each model can be found at https://github.com/meta-math/MetaMath?tab=readme-ov-file.

### B.2    INFORMATIVENESS AND MISSING STEP EXPLAINED

Both informativeness and missing step metrics are derived from the work of ROSCOE (Golovneva et al., 2023). Formally, given the problem descriptions as *source context* $s = \{s_1, ..., s_T\}$ of $T$-sentences, LLM generated solutions as *hypothesis* $h = \{h_1, ..., h_N\}$ of $N$-steps and ground truth solutions as *references* $r = \{r_1, ..., r_K\}$ of $K$-steps, informativeness and missing step are computed by measuring the semantics similarity among $s$, $h$ and $r$.

We select the Informativeness-Chain as the informativeness metric. This metric embeds the reasoning chain and source context as a whole is calculated by measuring the cosine similarity: Info-Chain $= [1 + \cos(h, s)]/2$, where we employ `all-mpnet-base-v2` as the sentence embedding model.

For missing step, we compute the similarity score between the reasoning steps from the ground truth and those generated by the LLMs to determine whether any steps are missing in the latter. Given each alignment value $\alpha_i = r\text{-align}(r_i \rightarrow h) = [1 + \max_{j=1}^{N}(\cos(r_i, h_j))] \in [0, 1]$ is the normalized cosine similarity between reference step and most similar step in the hypothesis, the missing step is defined as the minimal of similarity between each reference step and the hypothesis chain: Missing-Step $= \min_{i=1..K}(r\text{-align}(r_i \rightarrow h))$. Here we also employ `all-mpnet-base-v2` as the sentence embedding model to embed all the reasoning steps.

Furthermore, we conduct an error analysis on 50 randomly chosen bad cases in the fourth round of E-GSM and find that 46% (23/50) samples failed due to the incorrect extraction of known conditions and the remains are due to flawed reasoning paths.

### B.3    SFT DATA EXAMPLES

Our entire training set includes 64,929 CoT data, with 38,507 from the original CoT augmented training set $\mathcal{D}_0$ and 26,422 CoT data for extended questions $\mathcal{D}_1$. Following (Yu et al., 2023), we use the training prompt in Table 7, where the instruction is replaced by training example from $\mathcal{D}_0$ or $\mathcal{D}_1$. For scaling up extended questions, there are 24,147 training examples in $\mathcal{D}_2$ (see Section **??** and Appendix C.2). We showcase one concrete example from $\mathcal{D}_0$, $\mathcal{D}_1$, and $\mathcal{D}_2$ in Table 9 and one can find more examples in our released SFT dataset $\mathcal{D}$. $\mathcal{D}$ includes a total of approximately 89K CoT data, covering all examples within $\mathcal{D}_0, \mathcal{D}_1, \mathcal{D}_2$. This dataset will be made available under MIT License. Note that $\mathcal{D}_2$ is only used in Figure 7 and Appendix C.2.

### B.4    ANSWER EXTRACTION

Unlike previous works (Luo et al., 2023; Yu et al., 2023) that derive the final answer by extracting strings following the phrase "The answer is", we use GPT-3.5-turbo for answer extraction as given in Table 10. This strategy is adept at handling atypical cases that cannot be readily resolved using

Table 7: Training prompt for Open-source LLMs.

Table 8: Evaluation prompt for Open-source LLMs.

| |
|---|
| Below is an instruction that describes a task. Write a response that appropriately completes the request.

### Instruction:
{instruction}

### Response: |

| |
|---|
| Below is an instruction that describes a task. Write a response that appropriately completes the request.

### Instruction:
{instruction}

### Response: Let's think step by step |

Table 9: Training Examples.

| An example from $D_0$ |
|---|
| **Instruction**: Natalia sold clips to 48 of her friends in April, and then she sold half as many clips in May. How many clips did Natalia sell altogether in April and May?
**Response**: Natalia sold 48 clips in April. In May, she sold half as many, which is 48 / 2 = 24 clips. Altogether, she sold 48 + 24 = 72 clips in April and May. |
| **An example from $D_1$** |
| **Instruction**: In the bustling town of Willow Creek, Natalia had become quite the entrepreneur with her homemade clip business. In the month of April, her creativity and hard work paid off when she managed to sell her beautifully crafted clips to 48 of her friends, each one more eager than the last to support her venture. As the flowers began to bloom and May rolled around, Natalia continued her efforts, though she found that the demand had slightly decreased. By the end of May, she realized she had sold exactly half the number of clips she did in April. With these two months of sales combined, how many clips did Natalia sell altogether in April and May?
**Response**: In April, Natalia sold 48 clips. In May, she sold half of the number she sold in April, which is 48/2 = 24 clips. Therefore, Natalia sold 48 + 24 = 72 clips altogether in April and May. |
| **An example from $D_2$** |
| **Instruction**: In the charming town of Willow Creek, nestled among rolling hills and vibrant flower fields, Natalia had carved out a niche for herself as a budding entrepreneur with a unique talent for crafting homemade clips. Her designs, infused with creativity and a personal touch, quickly caught the eye of her friends and neighbors. In the blossoming month of April, as the community came alive with the sights and sounds of spring, Natalia's business flourished. She found herself busily fulfilling orders for 48 of her friends, each one more enthusiastic than the last to support her burgeoning venture and adorn their outfits with her unique creations. As the calendar turned to May, the fresh blooms of April gave way to the full splendor of spring. Natalia, ever dedicated to her craft, continued to pour her heart and soul into each clip she made. However, she noticed a slight shift in the air. The initial surge of excitement that had greeted her April launch had tempered somewhat, and the pace of orders had slowed. By the time the final days of May had drawn to a close, Natalia took stock of her efforts and realized that the number of clips she had sold this month was exactly half of what she had managed in April. Reflecting on these two months of hard work and the support of her community, Natalia wondered: How many clips had she sold altogether in April and May, marking the beginning of her journey as an entrepreneur in Willow Creek?
**Response**: In April, Natalia sold 48 clips to her friends. In May, she sold half of what she sold in April, which means she sold 48 / 2 = 24 clips in May. To find out how many clips she sold altogether in April and May, we add the number of clips sold in April and May: 48 (April) + 24 (May) = 72 clips. Therefore, Natalia sold 72 clips altogether in April and May, marking the beginning of her journey as an entrepreneur in Willow Creek. |

trigger words like "The answer is", which appears with various LLMs each presenting distinct output formats in zero-shot manner. Preliminary extraction has shown that over $99.8\%$ of responses have been accurately extracted. Following (Yu et al., 2023), for the SFT data mentioned in Section B.5, we append "The answer is" to the answers extracted from GPT-3.5-turbo at the end of responses. Nevertheless, to ensure fairness, we also employ our innovative answer extraction technique for our fine-tuned LLMs.

Table 10: Prompt for answer extraction.

---

Given the last 'Quesion' and 'Answer', your goal is to extract the final numerical result from the 'Answer' part, and put the numerical result after the 'Result' part. You must not to solve the problem by yourself, all you need to do is give me just a number extracted from 'Answer'.

---

### B.5 SFT DETAILS

In our experiments with the LLaMA-2 backbone, the learning rates are chosen based on the model scale. For LLMs with parameters of 7B and 13B, the learning rate is set to $0.00002$. For 70B LLMs, the learning rate is adjusted to $0.00001$. For the Mistral-7B base model, the learning rate is further reduced to $0.000005$ to maintain the stability of the training. The learning rate is set to $0.00002$ for LLaMA-3-8B and $0.00001$ for LLaMA-3-70B.

Batch sizes are also tailored to LLMs parameters to maximize the utilization of computational resources. Specifically, for the 70B model, we select a batch size of 24 per device. For models with 7B and 13B parameters, a larger batch size of 36 per device is chosen.

All models undergo training for 3 epochs with AdamW optimizer with a $3\%$ learning rate warmup. Experiments for LLMs with sizes 7B and 13B are conducted on $4 \times$ H800 GPUs. For the larger 70B model, necessitating more computational power, the experiments are carried out on $8 \times$ H800 GPUs (80G). The most computation-expensive experiment (fine-tune a 70B model) takes around 190 GPU hours. The entire experiment takes around 1,600 GPU hours (excluding preliminary or failed experiments).

During inference, we apply greedy decoding with a temperature of $0.0$ and the maximum generation length is set to $512$. Following (Yu et al., 2023), we use the zero-shot evaluation prompting, as shown in Table 8, where the instruction is replaced by the testing question.

All our SFT models will be made available for reproduction and future research endeavors. The license for these models will adhere to the same license applicable to the models prior to fine-tuning.

Multiple open-source LLMs, differing in scale and base models, are fine-tuned on $\mathcal{D}$ or $\mathcal{D}_0$. The fine-tuning is carried out over 3 epochs and the results are reported using 0-CoT with the vLLM library[4]. All evaluations adhere to the same 0-CoT instruction set to ensure consistency. To account for the potential influence of random variations during the training process, we include the performance trends throughout the entire training period in Appendix C.1. Furthermore, comprehensive details regarding the training and inference parameters can be found in Appendix B.5.

### B.6 CONTRAST COEFFICIENTS

$C = [C_1, C_2, ..., C_{k-1}]$ is the contrast coefficients, where each $C_i$ is assigned based on the hypothesized trend. The contrast $C$ is used to test the significance of the hypothesized trend across levels of $\beta$. In our case, the contrast coefficients that we use are centered linear trend coefficients. For example, $[-2. -1, 0, 1, 2]$ is used for $[C_1.C_2, ..., C_5]$. Note that the absolute difference $C_i - C_{i-1}$ does not matter because we want to test $C \cdot \beta = 0$ and we can divide both sides by any number except 0.

### B.7 GENERALIZABILITY SETUP

Here we give a brief introduction to the benchmark datasets we used in Section 4.5.

- MAWPS (Koncel-Kedziorski et al., 2016) is a benchmark of MWPs, incorporating 238 test examples. It is under MIT License and can be found at https://github.com/LYH-YF/MWPToolkit.
- SVAMP (Patel et al., 2021) includes 1,000 simple MWPs, which is available at https://github.com/LYH-YF/MWPToolkit. It is under MIT License.

---

[4]https://github.com/vllm-project/vllm.

- GSM-IC (Shi et al., 2023) is a variant of GSM8K, including MWPs with one irrelevant sentence. The dataset is available at https://github.com/google-research-datasets/GSM-IC. Following (Shi et al., 2023), we randomly sampled 4K questions as our test set for evaluation.

## C  FURTHER RESULTS

### C.1  TRAINING CURVES

Figure 6 depicts the CoLeG-E and CoLeG-R for both LLaMA-2-7B and Mistral-7B throughout 1,200 fine-tuning steps, equivalent to three complete epochs. Initially, the performance of both models improves rapidly before stabilizing at a later stage. This trend underscores the effectiveness of SFT in improving CoLeG.

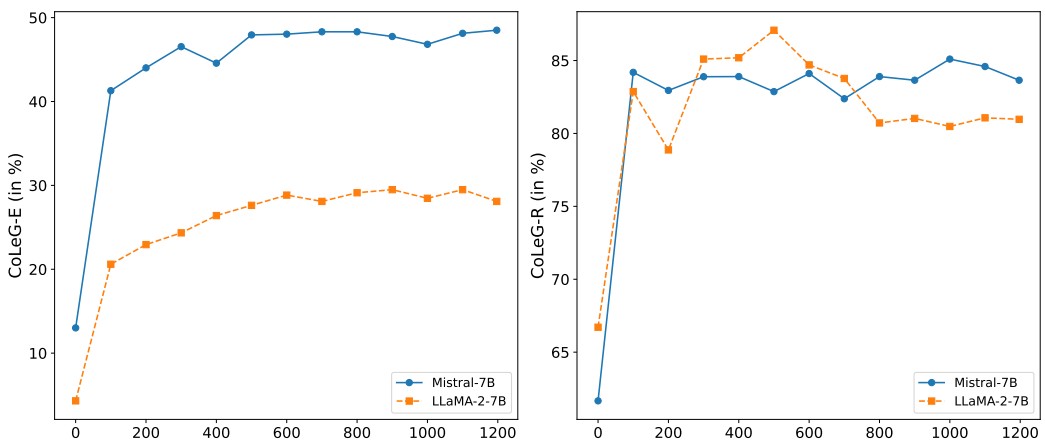

Figure 6: The performance curves of LLaMA-2-7B and Mistral-7B during 1,200 fine-tuning steps. Left: CoLeG-E; Right: CoLeG-R. The x-axis is the training steps during fine-tuning.

### C.2  SCALING UP SFT DATA

During the construction of the E-GSM (refer to Section 2.2), we use a round-by-round extension strategy, which can also be applied to augment the original GSM8K training set. The primary findings utilize $\mathcal{D} = \mathcal{D}_0 \cup \mathcal{D}_1$ for SFT, and we can study the effect of enlarging SFT dataset through this specific extension strategy based on $\mathcal{D}_1$. Consequently, we examine the effects of SFT on CoLeG in four distinct scenarios: without SFT, SFT on $\mathcal{D}_0$, SFT on $\mathcal{D}_0, \mathcal{D}_1$, SFT on $\mathcal{D}_0, \mathcal{D}_1, \mathcal{D}_2$. Figure have shown the performance in different settings, and the additional outcomes are presented in Figure 7 and 8.

### C.3  MORE RESULTS

We include additional resutls in Table 12. The results have shown the efficacy of our SFT method on specialized math LLMs.

## D  PROMPTS IN EXPERIMENTS

The 8-shot CoT experiment on GSM8K (see Section 2.1) with GPT-3.5-turbo (see Section B.1) uses prompts provided by (Wei et al., 2022). The full prompt is given in Table 13.

For the construction of the E-GSM (see Section 2.2), we provide the template for extension in Figure 2. Table 14 shows the complete prompts for the first two rounds of extension, and all prompts will be released in our code repository.

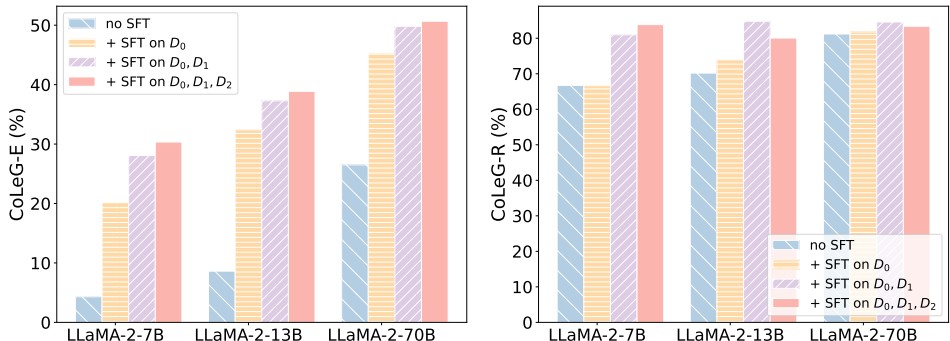

Figure 7: CoLeG-E (left) and CoLeG-R (right) of scaling up SFT dataset across various model scales in the LLaMA-2 family. The results suggest scaling up model scales and SFT dataset can further improve CoLeG.

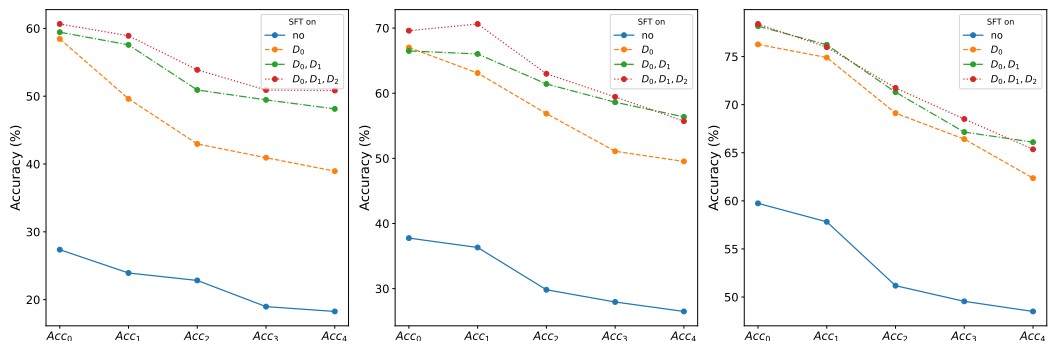

Figure 8: The changes in $Acc_i$ when LLMs are fine-tuned on different SFT dataset. Left: LLaMA-2-7B; Middle: LLaMA-2-13B; right: LLaMA-2-70B.

Moreover, all zero-shot prompting including our proposed prompt methods along with their format are presented in Table 15. Our investigated zero-shot prompting including zero-shot CoT (Kojima et al., 2022), Plan-and-Solve (Wang et al., 2023a) and its variant PS+.

**Zero-shot CoT** (Kojima et al., 2022), a variant of CoT (Wei et al., 2022), prompts the model directly with the problem of interest followed by "Let's think step by step," without providing any demonstration examples.

**Plan-and-Solve** (Wang et al., 2023a) enhances 0-CoT by initially prompting LLMs to create a plan that breaks down the overarching task into subtasks, subsequently executing these subtasks in accordance with the devised plan. Additionally, we investigate a variant of PS, identified as PS+, designed to mitigate calculation inaccuracies.

# E  CASE STUDY

We provide 2 solving cases of GPT-3.5-turbo with/without applying CoRe in Table 16. We could notice that faced with long context MWPs, 0-CoT fails to extract all the necessary conditions from the problems (with the missed condition highlighted in red), thus finally lead to wrong answers. In contrast, our CoRe enables LLMs to first focus on extracting conditions from the problem and then perform math reasoning based on these conditions, avoiding distractions from lengthy story details.

Table 11: Complete results of the effect of scaling up SFT dataset on CoLeG. All figures are in %. Differences between the SFT results and the original are shown in parentheses. The best results within each model scale are shown in bold. In general, scaling up SFT dataset can improve CoLeG.

| Model | CoLeG-E | CoLeG-R | $Acc_0$ | $Acc_1$ | $Acc_2$ | $Acc_3$ | $Acc_4$ |
|---|---|---|---|---|---|---|---|
| LLaMA-2-7B | 4.31 | 66.71 | 27.37 | 23.93 | 22.83 | 18.97 | 18.26 |
| + SFT on $\mathcal{D}_0$ | 20.22 (+15.91) | 66.64 (-0.07) | 58.45 (+31.08) | 49.62 (+25.69) | 42.96 (+20.13) | 40.93 (+22.96) | 38.95 (+20.69) |
| + SFT on $\mathcal{D}_0, \mathcal{D}_1$ | 28.09 (+23.78) | 80.97 (+14.26) | 59.44 (+32.07) | 57.57 (+33.64) | 50.92 (+28.09) | 49.46 (+30.49) | 48.13 (+29.87) |
| + SFT on $\mathcal{D}_0, \mathcal{D}_1, \mathcal{D}_2$ | **30.34** (+26.03) | **83.83** (+17.12) | **60.65** (+33.28) | **58.91** (+35.98) | **53.89** (+31.06) | **50.91** (+32.14) | **50.84** (+32.18) |
| LLaMA-2-13B | 8.61 | 70.18 | 37.76 | 36.32 | 29.83 | 27.95 | 26.50 |
| + SFT on $\mathcal{D}_0$ | 32.40 (+23.79) | 73.91 (+3.73) | 67.02 (+29.26) | 63.10 (+26.78) | 56.87 (+27.04) | 51.09 (+23.14) | 49.53 (+23.03) |
| + SFT on $\mathcal{D}_0, \mathcal{D}_1$ | 37.27 (+28.66) | **84.78** (+14.60) | 66.49 (+28.73) | 66.03 (+29.71) | 61.42 (+31.59) | 58.62 (+30.67) | **56.37** (+29.87) |
| + SFT on $\mathcal{D}_0, \mathcal{D}_1, \mathcal{D}_2$ | **38.86** (+30.25) | 80.05 (+9.87) | **69.60** (+31.84) | **70.63** (+34.31) | **62.99** (+33.16) | **59.44** (+31.49) | 55.71 (+29.21) |
| LLaMA-2-70B | 26.50 | 81.19 | 59.74 | 57.82 | 51.18 | 49.55 | 48.50 |
| + SFT on $\mathcal{D}_0$ | 45.32 (+18.82) | 81.76 (+0.57) | 76.27 (+16.53) | 74.90 (+17.08) | 69.12 (+17.94) | 66.42 (+16.87) | 62.36 (+13.86) |
| + SFT on $\mathcal{D}_0, \mathcal{D}_1$ | 49.81 (+23.31) | **84.57** (+3.38) | 78.17 (+18.43) | **76.23** (+18.41) | 71.30 (+20.12) | 67.15 (+17.60) | **66.10** (+17.60) |
| + SFT on $\mathcal{D}_0, \mathcal{D}_1, \mathcal{D}_2$ | **50.66** (+24.16) | 83.37 (+2.18) | **78.39** (+18.65) | 75.98 (+18.16) | **71.74** (+20.56) | **68.51** (+18.96) | 65.36 (+16.86) |

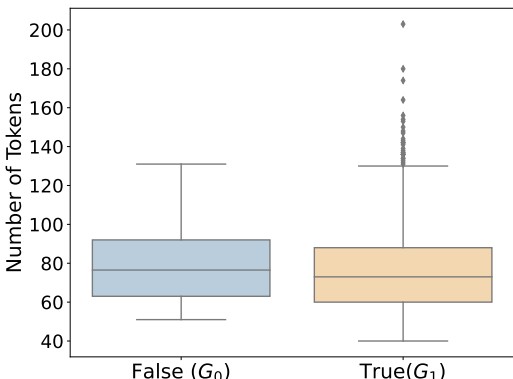

Figure 9: `GPT-4o-mini` also struggles to solve longer MWPs.

## F    ADDITIONAL ANALYSIS

As one reviewer suggested, we use `GPT-4o-mini` to do the same analysis as in Section 2.1. `GPT-4o-mini` achieves over 93% accuracy on GSM8K dataset, which a strong LLM. The results are illustrated in Figure 9, which have shown very similar trend as Figure 1. Following Section 2.1, there is significant evidence indicating that the number of tokens in $G_1$ is less than in $G_0$ with $U = 60462.5$, $P = 0.034$.

We also add the same analysis for `GPT-4o` and *OpenAI o1*, as shown in Figure 10 and 11. Following Section 2.1, for *GPT-4o* there is significant evidence indicating that the number of tokens in $G_1$ is less than in $G_0$ with $U = 45185$, $P = 0.0284$; for *OpenAI-o1*, the problem remains, although it is not that significant ($U = 47068.5, P = 0.1002$).

| Model | CoLeG-E | CoLeG-R | $Acc_0$ | $Acc_1$ | $Acc_2$ | $Acc_3$ | $Acc_4$ |
|---|---|---|---|---|---|---|---|
| LLaMA-2-7B Models | | | | | | | |
| MAmmoTH-7B | 11.52 | 66.09 | 47.46 | 45.02 | 34.91 | 32.67 | 31.37 |
| + SFT on $\mathcal{D}$ | 30.15 | 80.25 | 62.77 | 60.00 | 54.42 | 52.18 | 50.37 |
| | (+18.63) | (+14.16) | (+15.31) | (+14.98) | (+19.51) | (+19.51) | (+19.00) |
| WizardMath-7B | 11.89 | 59.72 | 52.99 | 43.77 | 38.41 | 33.58 | 31.65 |
| + SFT on $\mathcal{D}$ | 31.37 | 76.77 | 63.91 | 58.83 | 58.09 | 52.45 | 49.06 |
| | (+19.48) | (+17.05) | (+10.92) | (+15.06) | (+19.68) | (+18.87) | (+17.41) |
| MetaMath-7B | 31.74 | 74.13 | 66.57 | 62.76 | 54.51 | 51.27 | 49.34 |
| + SFT on $\mathcal{D}$ | 38.01 | 79.17 | 69.07 | 65.36 | 61.77 | 57.62 | 54.68 |
| | (+6.27) | (+5.04) | (+2.50) | (+2.60) | (+7.26) | (+6.35) | (+5.34) |
| Mistral-7B Models | | | | | | | |
| WizardMath-Mistral-7B | 50.94 | 79.13 | 82.71 | 76.90 | 70.78 | 68.78 | 65.45 |
| + SFT on $\mathcal{D}$ | 52.81 | 82.72 | 81.96 | 80.33 | 74.28 | 71.87 | 67.79 |
| | (+1.87) | (+3.59) | (-0.75) | (+3.43) | (+3.50) | (+3.09) | (+2.34) |
| MetaMath-Mistral-7B | 43.63 | 75.69 | 77.56 | 72.80 | 66.75 | 60.89 | 58.71 |
| + SFT on $\mathcal{D}$ | 49.72 | 82.33 | 78.24 | 77.32 | 71.13 | 68.78 | 64.42 |
| | (+6.09) | (+6.64) | (+0.68) | (+4.52) | (+4.38) | (+7.89) | (+5.71) |
| Other 7B Models | | | | | | | |
| llemma_7b | 2.06 | 66.63 | 28.81 | 27.64 | 23.10 | 20.05 | 19.19 |
| + SFT on $\mathcal{D}$ | 33.05 | 80.71 | 67.17 | 62.59 | 58.09 | 54.54 | 54.21 |
| | (+30.99) | (+14.08) | (+38.36) | (+34.95) | (+34.99) | (+34.49) | (+35.02) |
| deepseek-math-7b | 42.98 | 69.07 | 81.88 | 76.57 | 66.67 | 62.79 | 56.55 |
| + SFT on $\mathcal{D}$ | 51.31 | 80.34 | 81.35 | 77.91 | 72.35 | 67.88 | 65.36 |
| | (+8.33) | (+11.27) | (-0.53) | (+1.34) | (+5.68) | (+5.09) | (+8.81) |
| MetaMath-Llemma-7B | 28.18 | 64.77 | 68.23 | 62.34 | 53.81 | 49.55 | 44.19 |
| + SFT on $\mathcal{D}$ | 38.86 | 77.32 | 70.96 | 67.78 | 62.73 | 57.62 | 54.87 |
| | (+10.68) | (+12.55) | (+2.73) | (+5.44) | (+8.92) | (+8.07) | (+10.68) |
| LLaMA-2-13B Models | | | | | | | |
| MAmmoTH-13B | 18.63 | 74.07 | 55.12 | 52.97 | 45.76 | 45.01 | 40.82 |
| + SFT on $\mathcal{D}$ | 42.32 | 85.54 | 71.04 | 69.12 | 65.62 | 62.25 | 60.77 |
| | (+23.69) | (+11.47) | (+15.92) | (+16.15) | (+19.86) | (+17.24) | (+19.95) |
| WizardMath-13B | 19.29 | 72.12 | 57.77 | 54.90 | 45.49 | 42.29 | 41.67 |
| + SFT on $\mathcal{D}$ | 41.76 | 84.29 | 71.65 | 69.71 | 63.87 | 59.98 | 60.39 |
| | (+22.47) | (+12.17) | (+13.88) | (+14.81) | (+18.38) | (+17.69) | (+18.72) |
| MetaMath-13B | 36.14 | 75.68 | 71.27 | 68.20 | 60.45 | 55.99 | 53.93 |
| + SFT on $\mathcal{D}$ | 44.10 | 79.38 | 73.84 | 71.38 | 65.97 | 63.07 | 58.61 |
| | (+7.96) | (+3.70) | (+2.57) | (+3.18) | (+5.52) | (+7.08) | (+4.68) |
| LLaMA-2-70B Models | | | | | | | |
| WizardMath-70B | 45.97 | 78.75 | 80.14 | 76.49 | 69.29 | 65.34 | 63.11 |
| + SFT on $\mathcal{D}$ | 54.68 | 82.60 | 82.41 | 80.59 | 74.89 | 71.87 | 68.07 |
| | (+8.71) | (+3.85) | (+2.27) | (+4.10) | (+5.60) | (+6.53) | (+4.96) |
| MAmmoTH-70B | 42.98 | 85.24 | 72.93 | 74.48 | 69.20 | 64.52 | 62.17 |
| + SFT on $\mathcal{D}$ | 55.15 | 83.25 | 81.88 | 80.67 | 74.37 | 71.05 | 68.16 |
| | (+12.17) | (-1.99) | (+8.95) | (+6.19) | (+5.17) | (+6.53) | (+5.99) |
| MetaMath-70B | 52.81 | 80.86 | 82.34 | 77.57 | 71.92 | 67.79 | 66.57 |
| + SFT on $\mathcal{D}$ | 57.12 | 84.55 | 83.17 | 80.92 | 75.59 | 72.05 | 70.32 |
| | (+4.31) | (+3.69) | (+0.83) | (+3.35) | (+3.67) | (+4.26) | (+3.75) |

Table 12: Additional results of the effect of SFT on CoLeG. All figures are in %, and the SFT results are shaded. Differences between SFT results and original ones are shown in parentheses.

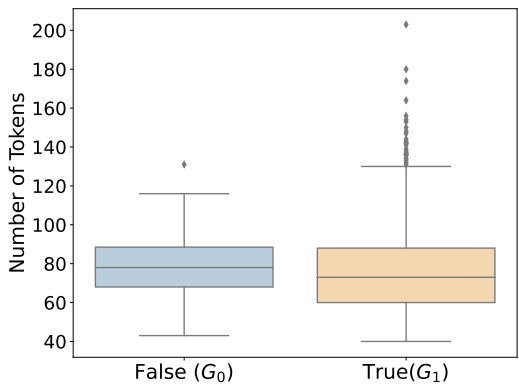

Figure 10: `GPT-4o` also struggles to solve longer MWPs.

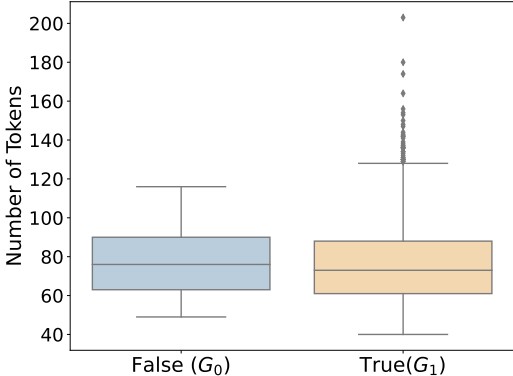

Figure 11: `OpenAI-o1` also struggles to solve longer MWPs, although the problem is not that significant than other models.

Table 13: 8-shot CoT Prompt.

**Q**: There are 15 trees in the grove. Grove workers will plant trees in the grove today. After they are done, there will be 21 trees. How many trees did the grove workers plant today?
**A**: There are 15 trees originally. Then there were 21 trees after the Grove workers planted some more. So there must have been 21 - 15 = 6 trees that were planted. The answer is 6.
**Q**: If there are 3 cars in the parking lot and 2 more cars arrive, how many cars are in the parking lot?
**A**: There are originally 3 cars. Then 2 more cars arrive. Now 3 + 2 = 5 cars are in the parking lot. The answer is 5.
**Q**: Leah had 32 chocolates and her sister had 42. If they ate 35, how many pieces do they have left in total?
**A**: Originally, Leah had 32 chocolates and her sister had 42. So in total they had 32 + 42 = 74. After eating 35, they had 74 - 35 = 39 pieces left in total. The answer is 39.
**Q**: Jason had 20 lollipops. He gave Denny some lollipops. Now Jason has 12 lollipops. How many lollipops did Jason give to Denny?
**A**: Jason had 20 lollipops originally. Then he had 12 after giving some to Denny. So he gave Denny 20 - 12 = 8 lollipops. The answer is 8.
**Q**: Shawn has five toys. For Christmas, he got two toys each from his mom and dad. How many toys does he have now?
**A**: Shawn started with 5 toys. He then got 2 toys each from his mom and dad. So he got 2 * 2 = 4 more toys. Now he has 5 + 4 = 9 toys. The answer is 9.
**Q**: There were nine computers in the server room. Five more computers were installed each day, from monday to thursday. How many computers are now in the server room?
**A**: There were originally 9 computers. For each day from monday to thursday, 5 more computers were installed. So 4 * 5 = 20 computers were added. Now 9 + 20 = 29 computers are now in the server room. The answer is 29.
**Q**: Michael had 58 golf balls. On tuesday, he lost 23 golf balls. On wednesday, he lost 2 more. How many golf balls did he have at the end of wednesday?
**A**: Michael started with 58 golf balls. He lost 23 on Tuesday, and lost 2 more on wednesday. So he had 58 - 23 = 35 at the end of Tuesday, and 35 - 2 = 33 at the end of wednesday. The answer is 33.
**Q**: Olivia has 23. She bought five bagels for 3 each. How much money does she have left?
**A**: Olivia had 23 dollars. She bought 5 bagels for 3 dollars each. So she spent 5 * 3 = 15 dollars. Now she has 23 - 15 = 8 dollars left. The answer is 8.
**Q**: {instruction}
**A**:

## G  BROADER IMPACTS

Our release of E-GSM presents a challenging benchmark to test the ability of LLMs to solve long MWPs. This aids future research by highlighting the intricacies involved in extended MWPs. Our innovative prompting method and the SFT technique have improved LLMs performance in these problems, potentially enhancing their generalization capabilities for other benchmarks. Note that both our E-GSM creation process and our methods can be easily adapted to other datasets or domains. This advancement not only broadens LLMs' mathematical reasoning capacities, but also has implications for educational tools. However, despite progress, significant challenges in solving extended MWPs remain, calling for future refinements. Additionally, there is a risk of overreliance which might impede the development of critical thinking skill and problem-solving skills for human learners. Therefore, we point out the need for balanced use of LLM technologies, combining their benefits with critical human oversight.

Table 14: Prompts to `gpt-4-turbo-preview` for extension.

| Round-1 |
| --- |
| Please expand the following math question into a story-like question with long context. Your modification of the question cannot change the meaning and the answer of the original question, and you should reply the new question without any problem solving steps and logical reasoning. You can not change the order of the original description.

**Original question**: If there are 3 cars in the parking lot and 2 more cars arrive, how many cars are in the parking lot?
**New question**: At 8:00 AM on a sunny Monday morning, the parking lot of the shopping mall was filling up with cars. Three sleek sedans were already parked in the designated spots. Among them are 1 red and 2 green. Their engines still warm from the short drive. 3 hours later, the sound of revving engines and honking horns could be heard as two more cars zoomed into the lot, eager to find a spot. The drivers quickly found a spot and joined the other three cars in the lot. With the addition of these two new arrivals, the number of cars in the lot had increased. How many cars were now parked in the lot, ready to be driven home by their owners after a long day of shopping?
**Original question**: There are 15 trees in the grove. Grove workers will plant trees in the grove today. After they are done, there will be 21 trees. How many trees did the grove workers plant today?
**New question**: At 10am, in the tranquil grove located in the outskirts of the bustling city, 5 workers arrived to plant trees. As they began their task, they noticed that there were currently 15 trees in the grove. Among them are 5 apple trees and 10 peach trees. With the sun shining down on them, they worked tirelessly for 4 hours, diligently planting new trees. By the time they finished, the grove was transformed with a total of 21 trees. How many trees were planted by the workers today?
**Original question**: {question to extend}
**New question**: |
| Round-2 |
| Please expand the following math question into a story-like question with longer context. Your modification of the question cannot change the meaning and the answer of the original question, and you should reply the new question without any problem solving steps and logical reasoning. You can not change the order of the original description.

**Original question**: At 8:00 AM on a sunny Monday morning, the parking lot of the shopping mall was filling up with cars. Three sleek sedans were already parked in the designated spots. Among them are 1 red and 2 green. Their engines still warm from the short drive. 3 hours later, the sound of revving engines and honking horns could be heard as two more cars zoomed into the lot, eager to find a spot. The drivers quickly found a spot and joined the other three cars in the lot. With the addition of these two new arrivals, the number of cars in the lot had increased. How many cars were now parked in the lot, ready to be driven home by their owners after a long day of shopping?
**New question**: At 11:00 AM, the shopping mall parking lot was bustling with activity. The sun was shining brightly, and the temperature was perfect for a day of shopping. As the clock struck 11:00, three sleek sedans were already parked in the designated spots, their engines still warm from the short drive. Among them were 1 red, 2 green. The owners of these cars had arrived early to beat the crowds and secure a good parking spot. However, as the morning went on, more and more cars began to fill up the lot. At 11:30 AM, a family in a minivan pulled into the lot, searching for a spot to park. They were followed shortly by a group of friends in a convertible, eager to start their day of shopping. With the addition of these two new arrivals, the number of cars in the lot had increased. How many cars were now parked in the lot, ready to be driven home by their owners after a long day of shopping?
**Original question**: At 10am, in the tranquil grove located in the outskirts of the bustling city, 5 workers arrived to plant trees. As they began their task, they noticed that there were currently 15 trees in the grove. Among them are 5 apple trees and 10 peach trees. With the sun shining down on them, they worked tirelessly for 4 hours, diligently planting new trees. By the time they finished, the grove was transformed with a total of 21 trees. How many trees were planted by the workers today?
**New question**: On a bright and sunny morning at 10:00 AM, in a serene grove situated on the outskirts of a lively city, a group of 5 dedicated workers gathered with a mission to enhance the greenery of the area. The grove, already home to 15 trees, boasted a mix of 5 apple trees and 10 peach trees, standing tall and providing shade and beauty to the surroundings. The workers, motivated by the pleasant weather and the peaceful environment, set out to plant additional trees, aiming to contribute to the grove's diversity and vibrancy. As the sun continued its journey across the sky, the workers labored for 4 hours, their hands working the soil and their spirits lifted by the task at hand. By 2:00 PM, their efforts had visibly paid off, and the grove now featured a total of 21 trees, each one promising growth and life. How many trees were planted by the workers on this productive day?
**Original question**: {question to extend}
**New question**: |

Table 15: All Zero-shot prompting methods and formats.

---

**Plan-and-Solve**
**Q**: { Question }
**A**: Let's first understand the problem and devise a plan to solve the problem. Then, let's carry out the plan to solve the problem step by step:

**Plan-and-Solve+**
**Q**: { Question }
**A**: Let's first understand the problem, extract relevant variables and their corresponding numerals, and devise a plan. Then, let's carry out the plan, calculate intermediate variables (pay attention to correct numeral calculation and commonsense), solve the problem step by step, and show the answer:

**Zero-shot CoT**
**Q**: { Question }
**A**: Let's think step by step:

**Condition-Retrieving Instruction**
**Q**: { Question }
**A**: **Let's first understand the problem, then list all the known conditions which are formed by numbers or quantitative relationships along with their contexts from problem text, and identify the final goal of the problem.** Let's think step by step:

---

Table 16: Math problems solving cases of `GPT-3.5-turbo` before / after applying CoRe.

| Problem Solving Case 1 from $Q_1$ |
|---|
| **Question:** Henry set out early in the morning on his trusty bicycle, determined to complete a 60-mile journey through the picturesque countryside. The sun was just beginning to rise, casting a golden glow over the fields and hills that lay ahead. After pedaling steadily for 20 miles, Henry decided to take his first break, stopping by a small stream to refill his water bottle and enjoy the peaceful sounds of nature. The journey was not yet halfway done, but he felt invigorated by the fresh air and the exercise. He knew that his second stop would be 15 miles before he reached his final destination, a quaint little café known for its homemade pies and fresh coffee. With this thought in mind, Henry resumed his journey, eager to discover how many miles lay between his first rest stop and the next. How many miles did Henry travel between his first and second stops during his scenic 60-mile bike trip? |
| **0-CoT Response:** Sure, let's break it down step by step: 1. Henry traveled 20 miles before taking his first break. 2. His second stop would be 15 miles before reaching his final destination. 3. To find out how many miles Henry traveled between his first and second stops, we need to subtract the distance of the second stop from the distance of the first stop. So, to calculate the distance between Henry's first and second stops: Distance between first and second stops = Distance of first stop - Distance of second stop. Distance between first and second stops = 20 miles - 15 miles. Distance between first and second stops = 5 miles. Therefore, Henry traveled 5 miles between his first and second stops. |
| **0-CoT + CoRe Response:** Sure, let's break down the problem and solve it step by step: 1. Known conditions: - Henry set out on a 60-mile journey. - He took his first break after pedaling 20 miles. - His second stop would be 15 miles before reaching his final destination. - The total distance of the journey is 60 miles. 2. Final goal: - Determine how many miles Henry traveled between his first and second stops. 3. Solution: - Henry took his first break after pedaling 20 miles, which means he has 60 - 20 = 40 miles left to travel. - His second stop would be 15 miles before reaching his final destination, so he traveled 40 - 15 = 25 miles from his first stop to his second stop. Therefore, Henry traveled 25 miles between his first and second stops during his scenic 60-mile bike trip. |

| Problem Solving Case 2 from $Q_1$ |
|---|
| **Question:** In a world where magic still whispers in the wind and mythical creatures roam the hidden corners of the Earth, there exists a rare and beautiful species known as unicorns. It is said that only 27 of these majestic creatures are left, safeguarding the last remnants of magic. Among these, a special group has made the misty Scottish Highlands their sanctuary, living amidst the rolling green hills and ancient forests. This secluded haven is home to one third of the world's remaining unicorns, where they roam freely, their manes glistening in the sunlight and their hooves barely touching the soft, mossy ground. Within this group, a remarkable two thirds are female, embodying the grace and strength of their kind. How many female unicorns are there among the Scottish unicorns, preserving the magic and mystery of their species for future generations to marvel at? |
| **0-CoT Response:** Sure, let's break it down step by step. 1. We know that two thirds of the unicorns in the Scottish Highlands are female. 2. Let's denote the total number of unicorns in the Scottish Highlands as "x". 3. Since two thirds of the unicorns are female, we can calculate the number of female unicorns as (2/3) * x. 4. Therefore, the number of female unicorns among the Scottish unicorns is (2/3) * x. If you have a specific number for the total unicorns in the Scottish Highlands, we can use that to calculate the number of female unicorns. |
| **0-CoT + CoRe Response:** Problem Understanding: - There are 27 unicorns in total in the world. - One third of the world's unicorns live in the misty Scottish Highlands. - Among the unicorns in the Scottish Highlands, two thirds are female. Known Conditions: - Total number of unicorns in the world: 27. - Number of unicorns in the Scottish Highlands: 27 / 3 = 9. - Percentage of female unicorns in the Scottish Highlands: 2/3. Final Goal: - Find the number of female unicorns among the Scottish unicorns. Solution: 1. Calculate the number of female unicorns in the Scottish Highlands: Number of female unicorns = Total number of unicorns in the Scottish Highlands * Percentage of female unicorns. Number of female unicorns = 9 * 2/3. Number of female unicorns = 6. Therefore, there are 6 female unicorns among the Scottish unicorns in the misty Scottish Highlands. |

