# OpenReview forum: "Can LLMs Solve Longer Math Word Problems Better?"
_ICLR.cc/2025/Conference — ICLR 2025 Poster_

### Official Review · Reviewer_cYhv · 2024-10-21

**Soundness:** 3
**Presentation:** 3
**Contribution:** 3
**Rating:** 6
**Confidence:** 4

**Summary:**

This paper investigates the performance of LLMs on Math Word Problems with extended narratives, introducing the concept of Context Length Generalizability (CoLeG). The authors created a new dataset, Extended Grade-School Math (E-GSM), by iteratively extending problems from GSM8K. They propose two novel metrics, CoLeG-E and CoLeG-R, to evaluate efficacy and robustness respectively. The study reveals that existing LLMs struggle with longer MWPs, showing a consistent performance decline as context length increases. To address this, the authors introduce Condition-Retrieving Instruction (CoRe) for proprietary LLMs and an extension-based fine-tuning approach for open-source LLMs. These methods demonstrate improvements in CoLeG across various LLM types and generalize well to other MWP benchmarks as well.

**Strengths:**

1. The paper addresses a gap in current research by focusing on LLMs' ability to handle longer MWPs, which is more reflective of real-world mathematical reasoning tasks. The focus on CoLeG provides insights into the limitations of current LLMs and pathways for improvement.

2. The creation of the E-GSM dataset through a systematic extension process is another contribution. By maintaining problem difficulty while increasing context length, the authors have developed a framework for evaluating LLM performance on longer MWPs.

3. The introduction of CoLeG-E and CoLeG-R metrics offers a more comprehensive evaluation framework than traditional accuracy measures. These metrics provide insights into both the consistency and robustness of LLM performance across varying context lengths.

4. The proposed methods, CoRe and extension-based fine-tuning, show consistent improvements across different LLM types and generalize well to other benchmarks.

**Weaknesses:**

1. The paper lacks a detailed exploration of why longer contexts impact LLM performance. While the authors mention potential working memory limitations, a deeper analysis could provide valuable insights. For instance, examining how performance correlates with the models' context window sizes or investigating the behavior of attention patterns in different layers could shed light on where breakdowns occur. Additionally, analyzing how different positional encoding schemes (e.g., rotary position embeddings vs. absolute position embeddings) affect performance on longer MWPs could offer insights into architectural considerations for improving CoLeG.

2. The E-GSM dataset creation process, while systematic, may introduce biases that aren't adequately addressed. Using GPT-4 for extensions could potentially lead to biases in language style, problem structure, or even subtle cues that GPT-4 uses for reasoning. For example, GPT-4 might consistently use certain phrases or sentence structures that inadvertently serve as hints for other GPT models. Additionally, there's a risk of amplifying any biases present in the original GSM8K dataset. The authors should consider analyzing the distribution of problem types, linguistic patterns, and solution strategies in E-GSM compared to the original dataset to identify any systematic biases introduced during extension.

3. The evaluation of open-source LLMs is limited to LLaMA-2 and Mistral-7B families. To provide a more comprehensive assessment, the authors should consider including models specifically designed for mathematical reasoning, such as MathGPT, GPT-f, or MetaMath. Additionally, evaluating performance on models with different architectural choices, like PaLM or BLOOM, could offer insights into how various model designs handle longer MWPs. This broader evaluation would strengthen the claims about the generalizability of the proposed methods.

4. While the paper shows improvements on other MWP benchmarks, it doesn't explore how the proposed methods perform on problems significantly longer than those in E-GSM. This leaves questions about the scalability of the approaches to even more complex, multi-page word problems. The authors could consider creating a small set of extremely long MWPs (e.g., 1000+ tokens) to test the limits of their methods and provide insights into scaling challenges.

5. The use of GPT-3.5-turbo for answer extraction in the evaluation process introduces a potential confounding factor. The paper doesn't adequately address how this might impact results, especially for non-OpenAI models. The authors should consider comparing this extraction method with simpler rule-based approaches or using model-specific output parsing to ensure fair comparison across different LLM families.

**Questions:**

1. How does the performance degradation on longer MWPs correlate with specific architectural features of different LLMs, such as context window size or attention mechanisms?

2. The extension approach shows promise for open-source LLMs. Have you considered how this might be adapted for extremely long MWPs or multi-step reasoning problems that span multiple pages?

---

> ### Author Response · Authors · 2024-11-19
> **Response to Reviewer cYhv**
>
> Dear Reviewer cYhv,
>
> Thank you for your time to review our work! Your feedback is really thoughtful and meaningful. We are sorry that we have tried our best and are only able to answer some of your questions as follows:
>
> > The paper lacks a detailed exploration of why longer contexts impact LLM performance. While the authors mention potential working memory limitations, a deeper analysis could provide valuable insights. For instance, examining how performance correlates with the models' context window sizes or investigating the behavior of attention patterns in different layers could shed light on where breakdowns occur. Additionally, analyzing how different positional encoding schemes (e.g., rotary position embeddings vs. absolute position embeddings) affect performance on longer MWPs could offer insights into architectural considerations for improving CoLeG.
>
> Thank you for your insightful comments! In fact, we have tried to investigate why longer contexts impact LLM performance and included a fine-grained analysis in Section 4.3, where we use two different metrics to capture the semantic understanding and missing steps in math reasoning. We find that both of these two are influenced by longer context.
>
> Thank you for pointing out alternative ways to analyze from attention patterns, positional encoding schemes, and context window sizes.
>
> - We believe analyzing attention patterns is interesting, but it is difficult. Even EMNLP best paper [1] only analyzes the pattern for classification tasks. The pattern and intrinsic attention mechanism is still open research questions. It is really a thoughtful comment and points to an interesting future work.
>
> - About the context windows. As the context window sizes are decided at the pretraining stage, we cannot afford to do such experiments. Additionally, if the input text is longer than the context window, the LLMs cannot “see” the tokens that exceed the max input length. However, our work focuses on the impact of the context if LLMs “see” the entire problem.
>
> - About positional encoding schemes. Similarly, positional encoding schemes are also decided at the pretraining stage. Additionally, we have incorporated LLMs with different positional encoding schemes in our experiments.
>
> > The evaluation of open-source LLMs is limited to LLaMA-2 and Mistral-7B families. To provide a more comprehensive assessment, the authors should consider including models specifically designed for mathematical reasoning, such as MathGPT, GPT-f, or MetaMath. Additionally, evaluating performance on models with different architectural choices, like PaLM or BLOOM, could offer insights into how various model designs handle longer MWPs. This broader evaluation would strengthen the claims about the generalizability of the proposed methods.
>
> Thank you for your suggestion! In Section 4.4, we include experiments with MetaMath. As suggested, we have included experiments about some specialized math LLMs in Appendix C.3 in our updated version.
>
> > The use of GPT-3.5-turbo for answer extraction in the evaluation process introduces a potential confounding factor. The paper doesn't adequately address how this might impact results, especially for non-OpenAI models. The authors should consider comparing this extraction method with simpler rule-based approaches or using model-specific output parsing to ensure fair comparison across different LLM families.
>
> Thank you for your question! We have already discussed this issue in Appendix B.4. The fact is that simple rule-based parsing is not accurate. For example, zero-shot-cot will use the last number as the final answer if they fail to parse the answer from the pattern “the answer is”. We randomly select 50 cases and find that the last number is not the answer for 9 out of 50 cases. There is no clear answer pattern for general-purpose LLMs (different from specialized math LLMs), that is why we use gpt-3.5-turbo to extract the answers. It is also a common practice to use LLMs as answer extractor [2].
>
> > The extension approach shows promise for open-source LLMs. Have you considered how this might be adapted for extremely long MWPs or multi-step reasoning problems that span multiple pages?
>
> Thank you for your recognition of our approach. Current MWPs are not that long and our SFT approach has shown great potential to create synthesized training data from short MWPs that is suitable for extremely long MWPs.
>
> We hope our response have addressed some of your concerns. We appreciate the opportunity to engage in discussion with a thoughtful reviewer like you. If you have any additional comments or would like to discuss further, please feel free to reach out to us.
>
>
> Sincerely,
>
> Authors
>
> [1] Label words are anchors: An information flow perspective for understanding in-context learning. arXiv preprint arXiv:2305.14160.
>
> [2] Mathvista: Evaluating mathematical reasoning of foundation models in visual contexts. arXiv preprint arXiv:2310.02255.

---

> > ### Comment · Reviewer_cYhv · 2024-11-26
> > **Reviewer Response**
> >
> > Thanks for the clarifications that you have made. I appreciate it.
> > While it does clear up some of the concerns, I believe the paper will significantly benefit from a thorough revision.
> > For the time being, I have decided to keep my scores.
> > Thanks.

---

> ### Author Response · Authors · 2024-11-27
>
> We truly appreciate the time and effort you’ve dedicated to reviewing our work.
>
> Based on your review, it appears that there are not many points requiring revision, except the point about "creating a small set of extremely long MWPs (e.g., 1000+ tokens)." We understand the value of investigating this scenario and considered this aspect during the rebuttal. However, we intentionally did not include such an experiment as we have anticipated Reviewer Dh39 will not buy this point either.
>
> We believe our approach has already demonstrated strong potential for tackling extremely long MWPs, as evidenced by the results from the E-GSM evaluation and generalization results in other MWP benchmarks. As highlighted in the manuscript, current MWP benchmarks do not include examples with very long contexts. Therefore, instead of manually constructing such benchmarks, we chose to focus on showcasing how our SFT-based approach can effectively generate synthesized training data from short MWPs while being well-suited for extending to tasks with longer contexts.
>
> May we kindly ask for additional suggestions to further strengthen our work? Your feedback in this regard would be incredibly valuable for us to improve our work.
>
> Thank you again for your insightful suggestions and for helping us further improve our study. We remain open to additional feedback and ways we can further improve.

---

### Official Review · Reviewer_jmci · 2024-10-21

**Soundness:** 3
**Presentation:** 3
**Contribution:** 2
**Rating:** 5
**Confidence:** 3

**Summary:**

This paper investigates the ability of LLMs to solve math word problems (MWPs) with longer contexts, introducing the concept of Context Length Generalizability (CoLeG). The key contributions are:(1) Creating Extended Grade-School Math (E-GSM), a dataset of MWPs with extended narratives. (2) Proposing two metrics to evaluate LLMs' efficacy and resilience on E-GSM. (3) Developing tailored prompts for proprietary LLMs to improve CoLeG. (4) Using extension as an auxiliary fine-tuning task for open-source LLMs. (5) Analyzing the impact on semantic understanding vs reasoning efficacy.

**Strengths:**

Strong motivation through rigorous statistical analysis shows LLMs struggle with longer MWPs (Section 2.1)

Proposes creative solutions (CoRe prompting and extension fine-tuning) to address identified limitations

Well-designed metrics (CoLeG-E and CoLeG-R) that capture both efficacy and robustness of LLMs on long MWPs

Sufficient experiments have proven the effectiveness of the method

**Weaknesses:**

The paper focuses on LLMs tackling longer math word problems, rather than genuinely difficult ones. Addressing truly challenging problems would likely yield more impactful and valuable research insights.

A deeper analysis of the types of errors LLMs make on extended MWPs would strengthen the paper. This could shed light on whether mistakes stem from misinterpreting context, losing track of key information, or actual computational errors.

The authors don't explore whether breaking down problems into atomic facts could help solve extended MWPs. It would be worthwhile to compare their methods against a baseline that first extracts crucial information from the lengthy context before attempting a solution. The techniques discussed in https://arxiv.org/abs/2305.14251 could be relevant here.

The table captions should be placed above the tables, not below, to comply with ICLR's official template guidelines.

The "Experimental Setup" section doesn't belong under Methodology. It should be moved to the Experiments section, alongside the results analysis.

**Questions:**

see weaknesses

---

> ### Author Response · Authors · 2024-11-18
> **Response to Reviewer jmci**
>
> Dear Reviewer jmci,
>
> Thank you for your time to review our work! We will answer your questions as follows:
>
> > The paper focuses on LLMs tackling longer math word problems, rather than genuinely difficult ones. Addressing truly challenging problems would likely yield more impactful and valuable research insights.
>
> Thank you for your concern! Our motivation stems from observing that even for grade school calculation problems, which are "not truly challenging", LLMs exhibit discrepancies in performance when faced with problems that have longer contexts (as discussed in Section 2.1). We find that the difficulty level of grade school MWPs is not a confounder of the impact of context length on math reasoning performance, as demonstrated in Section 2.1. We believe that the challenges posed by long contexts in MWPs represent a significant “challenging point” for current LLMs.
>
> Our study is centered on the influence of context length on LLM performance. To ensure a clear focus, we have deliberately isolated the effect of difficulty by controlling the difficulty level when creating E-GSM. We believe that pinpointing the current limitations of LLMs in solving MWPs could offer valuable insights and have a significant impact on future research and development.
>
> > A deeper analysis of the types of errors LLMs make on extended MWPs would strengthen the paper. This could shed light on whether mistakes stem from misinterpreting context, losing track of key information, or actual computational errors.
>
> Thank you for your suggestion! We have already included a deeper analysis to understand the underlying factors of performance decrease on E-GSM in Section 4.2 in the original manuscript (Section 4.3 in our updated version). As suggested, we have added an error analysis on 50 randomly chosen bad cases in the fourth round of E-GSM. We find that 46% (23/50) samples failed due to the incorrect extraction of known conditions and the remains are due to flawed reasoning paths. We have **included this part in Appendix B.2 in our updated manuscript**.
>
> > The authors don't explore whether breaking down problems into atomic facts could help solve extended MWPs. It would be worthwhile to compare their methods against a baseline that first extracts crucial information from the lengthy context before attempting a solution. The techniques discussed in https://arxiv.org/abs/2305.14251 could be relevant here.
>
> Thank you for pointing this out! We think this is not a baseline for the following reasons:
>
> - When we break down sentences into atomic facts, the context is still long or even longer.
>
> - They are totally different tasks. [1] uses this technique to do factual precision evaluation.
>
> - The idea might be in some sense similar to our CoRe.
>
> We have **included such discussion in Section 3.1 in our revised manuscript and cite this paper appropriately**:
>
> “[1] proposes a similar approach, suggesting that breaking down information into smaller components can enhance the evaluation of factual precision.”
>
> > The table captions should be placed above the tables, not below, to comply with ICLR's official template guidelines.
> > The "Experimental Setup" section doesn't belong under Methodology. It should be moved to the Experiments section, alongside the results analysis.
>
> Thank you for your comments! We have revised theses accordingly in our updated manuscript.
>
> We hope our response will address your concerns. If you have any further questions, feel free to discuss with us!
>
> Sincerely,
>
> Authors
>
> [1] Min, S., Krishna, K., Lyu, X., Lewis, M., Yih, W. T., Koh, P. W., ... & Hajishirzi, H. (2023). Factscore: Fine-grained atomic evaluation of factual precision in long form text generation. arXiv preprint arXiv:2305.14251.

---

### Official Review · Reviewer_Dh39 · 2024-10-29

**Soundness:** 2
**Presentation:** 1
**Contribution:** 2
**Rating:** 3
**Confidence:** 4

**Summary:**

In this paper, the authors investigated the performance of LLMs in solving long math problems. They first examined the performance discrepancy of ChatGPT 3.5 in solving two versions (i.e. long form v.s. short form) of the same questions and concluded that LLMs struggle to answer math word problems with longer context.

Then, they propose an automatic approach to extend the GSM questions into their long versions (named the obtained dataset E-GSM), with the same computation logic remaining, as far as they could.
After that, the paper presented a method called CoRe to help proprietary LLMs better handle these long-form questions.
For the open source LLMs, the authors fine-tuned them with a fine-tuning dataset comprising 65K CoT data, created by the authors.


The paper introduced E-GSM containing artificial long math problems, but in real cases, there are seldom questions written in the way that the authors presented, i.e. very verbose questions talking about a relatively simple math problem. Therefore, it is unknown whether the conduct here can help in solving real-world long math problems where although the question is quite long, it already describes the problem in a succinct way that it could. Better solving them is our goal, rather than solving the artificial verbose problems that are less likely to exist in the real world. Although they show the same characteristic length-wise, the capability of solving the latter is not necessarily helpful for solving the former.

**Strengths:**

The paper explored the impact of question length on LLMs’ performance and proposed a method to extend the length of GSM questions. The paper presented a method called CoRe to help proprietary LLMs better handle these long-form questions. For the open source LLMs, the authors fine-tuned them with a fine-tuning dataset comprising 65K CoT data, created by the authors.

**Weaknesses:**

1. The paper explores the artificial long math problems, but in real cases, there are seldom questions written in the way that the authors presented, i.e. very verbose questions talking about a relatively simple math problem. Therefore, it is unknown whether the conduct here can help in solving real-world long math problems where although the question is quite long, it already describes the problem in a succinct way that it could. Better solving them is our ultimate goal, rather than solving the artificial verbose problems that are less likely to exist in the real world.

2. Besides the above major point, there are more points:
- In Section 2.1, the authors examined the performance discrepancy of ChatGPT 3.5 in solving two versions (i.e. long form v.s. short form) of the same questions and concluded that LLMs struggle to answer math word problems with longer context. However, ChatGPT 3.5 is a relatively weak model now, I would suggest the authors do the same analysis with stronger open-source and proprietary LLMs.
- Still in Section 2.1, the analysis here is based on real math questions, but the long questions in E-GSM are artificial. Therefore, it is not convincing to me that the conclusion in Section 2.1 can provide a solid foundation for the subsequent conduct.

3. Many parts are not clear, see the questions section.

4. The writing needs a thorough improvement:
- “Human evaluation details are provided in Appendix A.4.” has a wrong reference.
- In the first paragraph of Section 3, the subsections should be introduced in order.
- The second sentence of Section 3.1 has redundancy.
- The first two sentences of Section 3.2 are not about open-source LLMs, therefore, they cannot help develop this section. The third sentence is redundant. In the fourth sentence, “their generated reasoning paths” should be referred to the place that telling how it is done. The loss function has a typo, should be “ (q, e, a)”.
- Section 3.3, “To negate the influence of few-shot demonstrations”, should be specific, what is the influence?
- Repeated sentences in the third paragraph of Section 4.1.

**Questions:**

1. According to “Evaluation results shows that 94.5% questions possess accepatable quality”, the total questions from rounds 1 to 4 should be about 5K. But in Table 1, it is only 4.5K.

2. As shown in Table 1, different rounds have different numbers of questions, what’s the impact on the defined metrics? namely CoLeG-E and CoLeG-R?

3. In Table 2, were the fine-tuned models evaluated with the CoRe method? can they be tested in the same way as those proprietary models?

4. “Apart from 7,473 annotated examples available in GSM8K training set, we get D0 that incorporate 38,507 valid CoT data points …”, the numbers here confused me. If the authors generated five reasoning paths for each question in the training set, at most, D0 can have 7,473*5 questions, less than 38,507.

5. In Section C.2, “The results suggest scaling up model scales and SFT dataset can further improve CoLeG.”, this conclusion may not be valid. Under CoLeG-R, after the SFT on D0, D1, and D2, the performance is not improved.

---

> ### Author Response · Authors · 2024-11-19
> **Response to Reviewer Dh39 (1/2)**
>
> Dear Reviewer Dh39,
>
> Thank you for your time to review our work! We will answer your questions as follows:
>
> > The paper explores the artificial long math problems, but in real cases, there are seldom questions written in the way that the authors presented, i.e. very verbose questions talking about a relatively simple math problem. Therefore, it is unknown whether the conduct here can help in solving real-world long math problems where although the question is quite long, it already describes the problem in a succinct way that it could. Better solving them is our ultimate goal, rather than solving the artificial verbose problems that are less likely to exist in the real world.
>
> Our research aims to investigate the effect of context length on math reasoning performance, specifically focusing on the inconsistencies observed when solving math word problems (MWPs) with longer contexts. We isolate the effect of intrinsic difficulty to ensure a clear understanding of how context length alone affects performance (as discussed in Section 2.1). By utilizing E-GSM and our devised metrics, we can effectively analyze how extending the context of the same problem impacts the performance of LLMs (refer to our analysis in Section 4.2).
>
> You might have overlooked some crucial aspects of our work that demonstrate the efficacy of our methods in addressing real-world MWPs:
>
> - Existing MWP benchmarks are not long, as highlighted in the caption of Table 3, which indicates that these benchmarks have fewer than 100 tokens. We are not introducing E-GSM as a new benchmark for long MWPs; rather, we are using it as a test ground to study our research question.
>
> - The results in Table 3 show that our method also provides benefits for solving real-world MWPs.
>
> - The analysis in Section 4.4 demonstrates that our method yields better improvements for relatively long questions in the GSM8K dataset.
>
>
> > In Section 2.1, the authors examined the performance discrepancy of ChatGPT 3.5 in solving two versions (i.e. long form v.s. short form) of the same questions and concluded that LLMs struggle to answer math word problems with longer context. However, ChatGPT 3.5 is a relatively weak model now, I would suggest the authors do the same analysis with stronger open-source and proprietary LLMs.
>
> The reason why we chose GPT-3.5-turbo is that it is cheap and efficient. Additionally, at the time we conduct our experiments, GPT-4o was not released. As suggested, we have added one strong model to do the same analysis in Appendix F in our revised manuscript.
>
>  > Still in Section 2.1, the analysis here is based on real math questions, but the long questions in E-GSM are artificial. Therefore, it is not convincing to me that the conclusion in Section 2.1 can provide a solid foundation for the subsequent conduct.
>
> Section 2.1 serves as the motivation for our work, as it highlights our finding that in GSM8K, LLMs struggle to solve relatively long problems effectively. We have specifically isolated the effect of difficulty level, demonstrating that problem context length is associated with degraded performance, which is a significant limitation of current LLMs. Our experiments on E-GSM confirm that when the context of the same problem is lengthened, LLM performance declines, consistent with the findings in Section 2.1. Building on this foundation, we propose different methods for both closed-source and open-source LLMs, demonstrating that our approaches are beneficial not only for E-GSM but also for some real-world MWPs. This underscores why Section 2.1 is a crucial foundation for our research.
>
>
> > The writing needs a thorough improvement:
> “Human evaluation details are provided in Appendix A.4.” has a wrong reference.
> In the first paragraph of Section 3, the subsections should be introduced in order.
> The second sentence of Section 3.1 has redundancy.
> The first two sentences of Section 3.2 are not about open-source LLMs, therefore, they cannot help develop this section. The third sentence is redundant. In the fourth sentence, “their generated reasoning paths” should be referred to the place that telling how it is done. The loss function has a typo, should be “ (q, e, a)”.
> Section 3.3, “To negate the influence of few-shot demonstrations”, should be specific, what is the influence?
> Repeated sentences in the third paragraph of Section 4.1.
>
> Thank you for pointing this out! We have revised these in our updated manuscript.

---

> ### Author Response · Authors · 2024-11-19
> **Response to Reviewer Dh39 (2/2)**
>
> > According to “Evaluation results shows that 94.5% questions possess accepatable quality”, the total questions from rounds 1 to 4 should be about 5K. But in Table 1, it is only 4.5K.
>
> As explained in Lines 173–176, we employ two heuristics to filter out "bad" extended questions. The specifics of these heuristics can be found in Appendix A.3, while the filtering process is detailed in Appendix A.4.
>
> The core idea behind our approach is to use entailment and solvability as metrics to filter out a substantial portion of questions, ensuring that all "bad" questions identified during our human evaluation are eliminated. This screening process explains why the number of questions presented in Table 1 diminishes with each successive round.
>
> > As shown in Table 1, different rounds have different numbers of questions, what’s the impact on the defined metrics? namely CoLeG-E and CoLeG-R?
>
> Please refer to Section 2.3 for the calculation process of our metrics. They are well-defined and specifically defined for different number of questions in each round. Additionally, they are fair for different LLMs.
> Additionally, CoLeG-E is defined according to the number of questions in the fourth round, and CoLeG-R is determined by accuracy in round 0 and round 4. As the number of questions of each round is larger than 1000, accuracy on each round is well-defined and statistically reliable.
>
> > In Table 2, were the fine-tuned models evaluated with the CoRe method? can they be tested in the same way as those proprietary models?
>
> No, they are not evaluated using the CoRe method. As detailed in Appendix B.5, we use the prompt specified in Table 8 for evaluation. They cannot be tested in the same manner as proprietary models because the evaluation prompt needs to be aligned with the training prompt, which is a common practice in the field [1, 2].
>
> > “Apart from 7,473 annotated examples available in GSM8K training set, we get D0 that incorporate 38,507 valid CoT data points …”, the numbers here confused me. If the authors generated five reasoning paths for each question in the training set, at most, D0 can have 7,473*5 questions, less than 38,507.
>
> As shown in Section 3.2, we filter out examples whose answers do not align with the ground truth. This process is referred to as RFT [3] and is widely adopted in the field [2, 3, 4].
>
> > In Section C.2, “The results suggest scaling up model scales and SFT dataset can further improve CoLeG.”, this conclusion may not be valid. Under CoLeG-R, after the SFT on D0, D1, and D2, the performance is not improved.
>
> CoLeG-R represents just one aspect of our evaluation. Both CoLeG-E and accuracy across all rounds have shown improvement.
>
>
> We hope our response will address your concerns. If you have any further questions, feel free to discuss with us!
>
> Sincerely,
>
> Authors
>
> [1] Luo, H., Sun, Q., Xu, C., Zhao, P., Lou, J., Tao, C., ... & Zhang, D. (2023). Wizardmath: Empowering mathematical reasoning for large language models via reinforced evol-instruct. arXiv preprint arXiv:2308.09583.
>
> [2] Yu, L., Jiang, W., Shi, H., Yu, J., Liu, Z., Zhang, Y., ... & Liu, W. (2023). Metamath: Bootstrap your own mathematical questions for large language models. arXiv preprint arXiv:2309.12284.
>
> [3] Yuan, Z., Yuan, H., Li, C., Dong, G., Lu, K., Tan, C., ... & Zhou, J. (2023). Scaling relationship on learning mathematical reasoning with large language models. arXiv preprint arXiv:2308.01825.
>
> [4] Tong, Y., Zhang, X., Wang, R., Wu, R., & He, J. (2024). Dart-math: Difficulty-aware rejection tuning for mathematical problem-solving. arXiv preprint arXiv:2407.13690.

---

> > ### Comment · Reviewer_Dh39 · 2024-11-21
> >
> > Thanks for your response, but unfortunately, the authors did not directly answer or solve most of my questions and concerns.
> >
> > For example:
> > - the gap between the generated verbose questions and the real-word long questions;
> > - why cannot use GPT4 for the experiment in this submission, given that the deadline for ICLR 25 is Oct 1st, 2024, far after the release of GPT4;
> > - ......
> >
> > I would suggest the authors respond again ASAP.

---

> > ### Comment · Reviewer_Dh39 · 2024-11-22
> >
> > >>“Apart from 7,473 annotated examples available in GSM8K training set, we get D0 that incorporate 38,507 valid CoT data points …”, the numbers here confused me. If the authors generated five reasoning paths for each question in the training set, at most, D0 can have 7,473*5 questions, less than 38,507.
> >
> > >As shown in Section 3.2, we filter out examples whose answers do not align with the ground truth. This process is referred to as RFT [3] and is widely adopted in the field [2, 3, 4].
> >
> > After the filtering, should it be less than 7,473*5?

---

> > > ### Author Response · Authors · 2024-11-22
> > >
> > > Thank you for your reply! Let us solve this one first.
> > >
> > > > After the filtering, should it be less than 7,473*5?
> > >
> > > As we mentioned in L288-L289: "Apart from 7,473 annotated examples available in GSM8K training set". We also incorporated the original training set, so the total number before filtering is 7473*6.

---

> > > > ### Comment · Reviewer_Dh39 · 2024-11-22
> > > >
> > > > But 7,473 is already the GSM8K training set, right? Which else is the original training set?

---

> ### Author Response · Authors · 2024-11-21
>
> Thank you for your response! We further explain as follows:
>
> **Q**: About the GPT4 experiments.
>
> **A**: As we have already explained, "The reason why we chose GPT-3.5-turbo is that it is cheap and efficient.". As you suggest, **we have already added the analysis of GPT-4o in Appendix F in our revised manuscript**.
>
> **Q**: About the gap between the generated verbose questions and the real-world long problems.
>
> **A**: You might have some misunderstandings! **Our E-GSM serves as a test bed of our research focus, and we have returned to the real-world problems in Table 3 and Section 4.4**. Our research aims to investigate the effect of context length on math reasoning performance, specifically focusing on the inconsistencies observed when solving math word problems (MWPs) with longer contexts. We isolate the effect of intrinsic difficulty to ensure a clear understanding of how context length alone affects performance (as discussed in Section 2.1). By utilizing E-GSM and our devised metrics, we can effectively analyze how extending the context of the same problem impacts the performance of LLMs (refer to our analysis in Section 4.2).
>
> You might have overlooked some crucial aspects of our work that demonstrate the efficacy of our methods in addressing real-world MWPs:
>
> - Existing MWP benchmarks are not long, as highlighted in the caption of Table 3, which indicates that these benchmarks have fewer than 100 tokens. We are not introducing E-GSM as a new benchmark for long MWPs; rather, we are using it as a test ground to study our research question.
>
> - The results in Table 3 show that our method also provides benefits for solving real-world MWPs.
>
> - The analysis in Section 4.4 demonstrates that our method yields better improvements for relatively long questions in the GSM8K dataset.
>
> If you have any questions, please drop on us, thank you!

---

> > ### Comment · Reviewer_Dh39 · 2024-11-22
> >
> > In your first response, it was mentioned that "we have added one strong model to do the same analysis in Appendix F in our revised manuscript.", which was GPT-4o-mini. I expected GPT-4o to be used, as we normally think GPT-4o has stronger reasoning capability than GPT-4o-mini. Moreover, I would like to see how GPT o1 performs here. In the new experiment with GPT-4o-mini in Figure 9, compared with Figure 1, the length gap between the False and True groups becomes much smaller.
> >
> > In Table 3, the average number of tokens of MAWPS, SVAMP, and GSM-IC are 52, 54, 80, respectively. However, in your E-GSM, the average length of Q_1 questions is about 192, and more than 300 from Q_2. I do not think there is a strong rationale to believe it solved my original concerns.

---

> > > ### Author Response · Authors · 2024-11-22
> > >
> > > Thank you for your engagement in further discussion!
> > >
> > > >  I expected GPT-4o to be used, as we normally think GPT-4o has stronger reasoning capability than GPT-4o-mini. Moreover, I would like to see how GPT o1 performs here.
> > >
> > > Sorry about the confusion. The reason we use GPT4o-mini is: 1. It already achieves over 93% in GSM8K, which is strong on GSM8K. 2. It is cheaper than GPT4o and in the same series.
> > >
> > > As you suggested, we are now adding GPT-4o and o1. We will let you know when it is done.
> > >
> > >
> > > > In the new experiment with GPT-4o-mini in Figure 9, compared with Figure 1, the length gap between the False and True groups becomes much smaller.
> > >
> > > Even for a strong model like GPT-4o-mini (on GSM8K, over 93% accuracy), **there is still a gap between the False and True groups**, which is indeed a limitation in math reasoning (under statistical significance).
> > > The interesting thing is that **this phenomenon aligns well with our results in Table 2 and analysis in Section 4.2**, the stronger LLM tends to have better performance on E-GSM in our metrics, which also **justifies the reasonability of our design principal of E-GSM**.
> > > We believe the usefulness of E-GSM is to enlarge this performance gap among different LLMs that may be ignored in the original GSM8K and show the rankings of LLMs in this perspective.
> > >
> > >
> > >
> > > > In Table 3, the average number of tokens of MAWPS, SVAMP, and GSM-IC are 52, 54, 80, respectively. However, in your E-GSM, the average length of Q_1 questions is about 192, and more than 300 from Q_2. I do not think there is a strong rationale to believe it solved my original concerns.
> > >
> > > I think **the main logic of our work is**: we discern a limitation in Section 2.1, develop a test bed to discuss and investigate this problem (build up E-GSM), propose our methods and show their efficacy in E-GSM, then we return to real-world problems (Table 3 and Section 4.2)
> > > **Additional evidence could be the second paragraph (Lines 453-463) in Section 4.2 and Figure 5 (right)**, where our method can improve the performance of relatively longer real-world problems in GSM8K (83-203 tokens).
> > > Another reason is that the current MWP benchmarks are somewhat short, and that is why we develop a new test bed to investigate our research problem.

---

> > > ### Author Response · Authors · 2024-11-23
> > >
> > > Our manuscript is updated to incorporate GPT4o and o1 in Appendix F.
> > >
> > > The results also show the context length is still a problem for these strong models.

---

> > > > ### Comment · Reviewer_Dh39 · 2024-11-26
> > > >
> > > > Thanks for the update.
> > > >
> > > > My major concern remains, i.e. the discrepancy in characteristics of your artificial long questions and the real natural long questions, and the question length difference in the real testbeds and your E-GSM. They are extremely important because you are studying "Can LLMs Solve Long Math Word Problems Better?", I do expect the conduct should involve real and natural long math questions.

---

> > > > > ### Author Response · Authors · 2024-11-27
> > > > >
> > > > > Thank you for your reply.
> > > > >
> > > > > We still believe your major concern is not a problem, the reasons are as follows:
> > > > >
> > > > > - Our title should be "Can LLMs Solve Longer Math Word Problems Better" (in PDF), which highlight out research focus: investigating the effect of longer context on solving MWPs. E-GSM serves as a good testbed as it isolates the effect of difficulty level and check the performance discrepancy of the same problems with longer and longer context.
> > > > >
> > > > > - About the discrepancy of characteristics.The main results have shown the performance drop of LLMs in E-GSM, which aligns well with the real-world GSM8K (in Section 2.1, which also serves as our motivation). Results of our methods improve the performance in E-GSM and also show superior results in many real-world MWP benchmarks. We believe these two points have shown the reliability and reasonability of using E-GSM.
> > > > >
> > > > > - About the question length difference. The second paragraph (Lines 453-463) in Section 4.2 and Figure 5 (right) have already shown that our method can improve the performance of relatively longer real-world problems in GSM8K (83-203 tokens). Additionally, current MWP benchmarks do not include examples with very long contexts. Introducing such an benchmark could deserve a paper in "dataset and benchmark" track. That is why we resort to transforming existing benchmarks (GSM8K) to get a new testbed. We are studying a limitation of current LLMs, not releasing a benchmark. We believe our work is a good trail in this direction. It is also unreasonable to require results on benchmarks with the exact same tokens as E-GSM, as existing MWPs benchmarks are not that long. Additionally, we have revealed that the main improvement of our approach is from "our methods could improve the performance of relatively longer MWPs", which also aligns well with the word in our title "longer".

---

> > > > > > ### Comment · Reviewer_Dh39 · 2024-11-27
> > > > > >
> > > > > > A quick question first. If "It is also unreasonable to require results on benchmarks with the exact same tokens as E-GSM, as existing MWPs benchmarks are not that long.", what's the point of make such long and verbose questions in E-GSM?

---

> ### Author Response · Authors · 2024-11-22
>
> Yes, 7473 is already the training set. "The original training set" refers to the GSM8K training set. Then we also generate 7473 * 5 and filter some wrong answers. The total number of questions after filtering should be less than 7473 + 7473*5 (we filter some in this part). By the way,  38,507 >  37,365 = 7473 *5.
>
> To elaborate more, 38507 - 7473 (the GSM8K training set) = 31,034, which is our generated new samples after filtering. We filter 7473*5 - 31,034 = 6,331 examples whose answers are wrong.
>
> We **have updated this sentence in our revised manuscript** to make it more accurate (L288-289 in our new version). Sorry for the confusion!

---

> ### Author Response · Authors · 2024-11-27
>
> Thank you for your question!
>
> We have already mentioned the main focus of E-GSM:
>
> - "Our research aims to investigate the effect of context length on math reasoning performance, specifically focusing on the inconsistencies observed when solving math word problems (MWPs) with longer contexts. We isolate the effect of intrinsic difficulty to ensure a clear understanding of how context length alone affects performance (as discussed in Section 2.1)."
>
> - "Introducing such an benchmark could deserve a paper in "dataset and benchmark" track. That is why we resort to transforming existing benchmarks (GSM8K) to get a new testbed. We are studying a limitation of current LLMs, not releasing a benchmark."
>
> Additional reasons:
>
> - As existing MWPs benchmarks are not that long, how could we test the performance on long MWPs? That is why we resort to transforming existing benchmarks (GSM8K) to get a new testbed. We are studying a limitation of current LLMs, not releasing a benchmark.
>
> - By your point, if a benchmark not exists, there is no need to study this field? In fact, there are many endeavor [1, 2] that adapt existing benchmarks to study specific research questions. They are no real math problems occurring in the way in [1, 2]. However, it is still worthwhile to do so because we expect our LLMs becomes stronger and stronger and could handle any unreal case. Another unreal case should be [3]. One of the reasons to conduct these is to inspect LLMs' ability from different aspects. Our research falls into this category.
>
> [1] https://huggingface.co/datasets/reasoning-machines/gsm-hard
>
> [2]  Large Language Models Can Be Easily Distracted by Irrelevant Context. ICML 2023. https://arxiv.org/abs/2302.00093
>
> [3] https://github.com/gkamradt/LLMTest_NeedleInAHaystack

---

> > ### Comment · Reviewer_Dh39 · 2024-11-28
> >
> > > GSM-hard: "We construct this dataset by replacing the numbers in the questions of GSM8K with larger numbers that are less common."
> >
> > So it is mostly real math questions, the question description is natural.
> >
> >
> > > Scaling Relationship on Learning Mathematical Reasoning with Large Language Models
> >
> > Not a peer-reviewed paper. Moreover, I cannot get its resemblance with your work.
> >
> >
> > > Needle in the Haystack for Memory Based Large Language Models
> >
> > Not a peer-reviewed paper.

---

> > > ### Author Response · Authors · 2024-11-28
> > >
> > > By your point, we should also find approximately the same number in the question and there is discrepancy between GSM-hard and real-world scenario as no humans will encounter such problems in their life.
> > >
> > >
> > > The second reference is [2] Large Language Models Can Be Easily Distracted by Irrelevant Context. ICML 2023
> > >
> > > > Needle in the Haystack for Memory Based Large Language Models not a peer-reviewed paper.
> > >
> > > This test is run by GPT-4, Claude, and many other famous LLMs [1, 2], which has shown its usefulness. It is also artificial case, the point is to test LLMs' capability from various facets.
> > >
> > > [1] Liu, A., Feng, B., Wang, B., Wang, B., Liu, B., Zhao, C., ... & Xu, Z. (2024). Deepseek-v2: A strong, economical, and efficient mixture-of-experts language model. arXiv preprint arXiv:2405.04434.
> > >
> > > [2] Yang, A., Yang, B., Hui, B., Zheng, B., Yu, B., Zhou, C., ... & Fan, Z. (2024). Qwen2 technical report. arXiv preprint arXiv:2407.10671.

---

> > ### Comment · Reviewer_Dh39 · 2024-11-28
> >
> > The original second reference was "Scaling Relationship on Learning Mathematical Reasoning with Large Language Models". I was not notified your edits of changing this reference to "Large Language Models Can Be Easily Distracted by Irrelevant Context."

---

> > > ### Author Response · Authors · 2024-11-28
> > >
> > > Additional reasons:
> > >
> > > - As existing MWPs benchmarks are not that long, how could we test the performance on long MWPs? That is why we resort to transforming existing benchmarks (GSM8K) to get a new testbed. We are studying a limitation of current LLMs, not releasing a benchmark.
> > >
> > > - By your point, if a benchmark not exists, there is no need to study this field? In fact, there are many endeavor [1, 2] that adapt existing benchmarks to study specific research questions. They are no real math problems occurring in the way in [1, 2]. However, it is still worthwhile to do so because we expect our LLMs becomes stronger and stronger and could handle any unreal case. Another unreal case should be [3]. One of the reasons to conduct these is to inspect LLMs' ability from different aspects. Our research falls into this category.
> > >
> > >
> > > Do you receive notification this time?
> > >
> > > [1] https://huggingface.co/datasets/reasoning-machines/gsm-hard
> > >
> > > [2] Large Language Models Can Be Easily Distracted by Irrelevant Context. ICML 2023. https://arxiv.org/abs/2302.00093
> > >
> > > [3] https://github.com/gkamradt/LLMTest_NeedleInAHaystack

---

> > > ### Author Response · Authors · 2024-11-28
> > >
> > > By the way, the edit time is before your response time. If you complain about there is no editing notification, why not email PCs about this issue?

---

### Official Review · Reviewer_eQsN · 2024-11-05

**Soundness:** 3
**Presentation:** 3
**Contribution:** 2
**Rating:** 5
**Confidence:** 4

**Summary:**

This work examines the effect of extended contexts on mathematical reasoning and introduces the Extended Grade-School Math (E-GSM) dataset, featuring math problems with lengthy narratives. Analysis reveals that current LLMs struggle with E-GSM, prompting the authors to propose new methods to address these challenges.

For proprietary LLMs, they introduce a new instructional prompt, while for open-source LLMs, they develop a novel auxiliary fine-tuning task. These approaches aim to enhance model performance in handling extended-context MWPs.

**Strengths:**

- This paper introduces E-GSM, a dataset with lengthy, distracting sentences that make it considerably more challenging than the original GSM. This dataset offers a valuable tool for evaluating the robustness of LLMs.

- The approach used to create E-GSM can also be applied to expand existing math training datasets, providing new supervised fine-tuning (SFT) data in the math domain.

**Weaknesses:**

- The augmented math questions may include contradicting sentences. The augmented math questions  may become unsolvable or yield answers that differ from the original ones.
Although human evaluations on 200 samples suggest that “94.5% of questions meet acceptable quality,” this accuracy may still be inadequate, particularly given that the labels in the GSM8K test set might contain errors.
An alternative could be to release these 200 samples as a verified subset of the E-GSM dataset. Reporting CoLeG-E and CoLeG-R results on the 200 samples, both with and without verification, would also be helpful.

- In Table 2, the higher results w/ $\mathcal{D}$ (compared to w/ $\mathcal{D_0}$) may because the size of  $\mathcal{D}$ is larger than $\mathcal{D_0}$.

**Questions:**

- How is E-GSM different from GSM-IC[1]?

[1] Large Language Models Can Be Easily Distracted by Irrelevant Context. ICML 2023. https://arxiv.org/abs/2302.00093

---

> ### Author Response · Authors · 2024-11-18
> **Response to Reviewer eQsN**
>
> Dear Reviewer eQsN,
>
> Thank you for your time to review our work! We will answer your questions as follows:
>
> > The augmented math questions may include contradicting sentences. The augmented math questions may become unsolvable or yield answers that differ from the original ones. Although human evaluations on 200 samples suggest that “94.5% of questions meet acceptable quality,” this accuracy may still be inadequate, particularly given that the labels in the GSM8K test set might contain errors. An alternative could be to release these 200 samples as a verified subset of the E-GSM dataset. Reporting CoLeG-E and CoLeG-R results on the 200 samples, both with and without verification, would also be helpful.
>
> Thank you for your question. The human evaluation criteria are detailed in Appendix A.2. Specifically, any question that includes contradictory sentences or yields a different answer from the original problem is classified as "poor" quality. **As explained in Lines 173–176**, we employ two heuristics to filter out "bad" extended questions. The specifics of these heuristics can be found in Appendix A.3, while the filtering process is detailed in Appendix A.4. The core idea behind our approach is to use entailment and solvability as metrics to filter out a substantial portion of questions, ensuring that all "bad" questions identified during our human evaluation are eliminated. This screening process explains why the number of questions presented in Table 1 diminishes with each successive round.
>
>
> > In Table 2, the higher results w/D (compared to w/$D_0$) may because the size of D is larger than $D_0$.
>
> Thank you for pointing this out. We expand $D_0$ to the same size of D by further RFT [1]. The results for Llama-2-7B is given as follows:
>
> | Method | CoLeG-E | CoLeG-R | $Acc_0$ | $Acc_1$ | $Acc_2$ | $Acc_3$ | $Acc_4$ |
> | - | - | - | - | - | - | - | - |
> | $D_0$ | 20.22 | 66.64 | 58.45 | 49.62 | 42.96 | 40.94 | 38.95 |
> | expanded $D_0$ | 20.34 | 66.28 | 58.99 | 50.06 | 43.35 | 41.25 | 39.10 |
> | $D_1$ | **28.09** | **80.97** | **59.44** | **57.57** | **50.92** | **49.44** | **48.13** |
>
> We can see that there is not much improvement and the claim remains. We hypothesize that this is because the set of unique questions remains unchanged; simply applying RFT yields similar solutions, resulting in minimal improvement for SFT. Moreover, the addition of short questions does not substantially enhance performance for E-GSM.
>
>
> > How is E-GSM different from GSM-IC?
>
> Thank you for your question! E-GSM is different with GSM-IC in the following way:
>
> - E-GSM is more challenging (not in terms of difficulty level of problems) than GSM-IC. GSM-IC uses template-based method to insert one irrelevant sentence to GSM8K problems, which initially reduced the performance of earlier LLMs like text-davinci-003. However, as LLMs become more sophisticated, GSM-IC no longer poses a significant challenge. For example, the current version of GPT-3.5-turbo achieves 88.35% accuracy in GSM-IC with 0-CoT (as shown in Table 3). In contrast, our E-GSM extends the context of GSM8K problems to create longer scenarios, which are inherently more challenging. Specifically, the accuracy of GPT-3.5-turbo on the fourth round of E-GSM is only 64.42% with 0-CoT.
>
> - Different research focus. GSM-IC explores the impact of introducing a single irrelevant sentence on the mathematical reasoning capabilities of LLMs. In contrast, our research with E-GSM is intended to examine the inconsistency of LLMs when solving extended math problems of the same difficulty level, as motivated by our discussion in Section 2.1.
>
> We hope our response will address your concerns. If you have any further questions, feel free to discuss with us!
>
> Sincerely,
>
> Authors
>
>
> [1] Yuan, Z., Yuan, H., Li, C., Dong, G., Lu, K., Tan, C., ... & Zhou, J. (2023). Scaling relationship on learning mathematical reasoning with large language models. arXiv preprint arXiv:2308.01825.
>
>
> .

---

### Meta-Review · Area_Chair_sZqg · 2024-12-22

**Metareview:**

This paper presents E-GSM, a collection of math word problems that feature lengthy narratives and then propose two novel metrics to evaluate whether current LLMs can handle these problems.  They evaluate several proprietary LLMs and some open source LLMs to see how they perform on this collection.  They also fine tune the open source models to perform better on these tasks.

Strengths:
1.  Contribution of a new dataset.
2.  Analysis of various LLMs on longer MWPs.
3.  Significant number of experiments.
4.  New metrics:   CoLeG-E and CoLeG-R.

Weaknesses:
1.  More deeper analysis of why LLMs have issues with such longer MWPs.
2. Answer extraction using GPT3.5 (there has been discussion about this).
3.  There are some writing improvement suggestions.

**Additional Comments On Reviewer Discussion:**

There has been a lot of discussion between the authors and the reviewers for this paper.  I decided to ignore the review of Dh39 since I did not find the points very pertinent to a fair evaluation of the paper.

---

### Decision · Program_Chairs · 2025-01-22

Accept (Poster)